# Source apportionment of black carbon aerosols from light absorption observation and source-oriented modeling: An implication in a coastal city in China

Junjun Deng[1], Hao Guo[2], Hongliang Zhang[3], Jialei Zhu[1], Xin Wang[1], Pingqing Fu[1]

[1] Institute of Surface-Earth System Science, School of Earth System Science, Tianjin University, Tianjin 300072, China

[2] Department of Earth System Science, University of California Irvine, CA 92697-3100, USA

[3] Department of Environmental Science and Engineering, Fudan University, Shanghai 200438, China

*Correspondence to:* Junjun Deng (dengjunjun@tju.edu.cn), Hongliang Zhang (zhanghl@fudan.edu.cn)

**Abstract.** Black carbon (BC) is the most important light absorbing aerosol in the atmosphere. However, sources of atmospheric BC aerosols are largely uncertain, making it difficult to assess its influence on radiative forcing and climate change. In this study, year-round light-absorption observations were conducted during 2014 using an Aethalometer in Xiamen, a coastal city in southeast China. Source apportionment of BC was performed and temporal variations in BC sources were characterized based on both light absorption measurements and a source-oriented air quality model. The annual average concentrations of BC from fossil fuel ($BC_{ff}$) and biomass burning ($BC_{bb}$) by the Aethalometer method were $2932 \pm 1444$ ng m$^{-3}$ and $1340 \pm 542$ ng m$^{-3}$, contributing 66.7 % and 33.3 % to total BC, respectively. A sensitivity analysis was performed with different Absorption Ångström Exponent (AAE) values of fossil fuel combustion ($\alpha_{ff}$) and biomass burning ($\alpha_{bb}$), suggesting that the Aethalometer method was more sensitive on changes in $\alpha_{bb}$ than $\alpha_{ff}$. $BC_{bb}$ contribution exhibited clear diurnal cycle with the highest level (37.9 %) in the evening rush hour and seasonal pattern with the maximum (39.9 %) in winter. Conditional probability function (CPF) analysis revealed the large biomass burning contributions were accompanied by east-northeasterly and northerly winds. Backward trajectory indicated that air masses from north and east-central China were associated with larger biomass burning contributions. Potential source contribution function (PSCF) and concentration-weighted trajectory (CWT) suggested that north and east-central China and Southeast Asia were potential sources for both $BC_{ff}$ and $BC_{bb}$. The source-oriented modeling results showed that transportation, residential and open biomass burning accounting for 45.3 %, 30.1 % and 17.6 % were the major BC sources. Among the three fuel catalogs, liquid fossil fuel (46.5 %) was the largest source, followed by biomass burning (32.6 %) and coal combustion (20.9 %). Source contributions of fossil fuel combustion and biomass burning identified by the source-oriented model were 67.4 % and 32.6 %, respectively, close to those obtained by the Aethalometer method. The findings provide solid supports for controlling fossil fuel sources to limit the impacts of BC on climate change and environmental degradation in the relatively clean region in China.

# 1 Introduction

Black carbon (BC) aerosol is a vital air pollutant throughout the surface earth system and it has attracted a great concern regarding its multiple impacts on human health, climate change and atmospheric visibility (Bond et al., 2013; Zhuang et al., 2018, 2019; Chen et al., 2020). As the most important light-absorbing component of PM$_{2.5}$ (particulate matters with aerodynamic diameter less than 2.5 µm), BC exerts a key and unique role in the climate system by absorbing solar radiation, affecting chemical/physical properties of cloud, and influencing snow and ice cover (Jacobson, 2002; Ramanathan and Carmichael, 2008; Bond et al., 2013; Qian et al., 2014; Kim et al., 2015). BC is even found as the second most important climate-warming agent after carbon dioxide, with a positive climate forcing of 1.1 W m$^{-2}$, greater than that of methane (Bond et al., 2013). BC also has impacts on urban weather conditions and may play a key role in extreme weather (Ding et al., 2013; Fan et al., 2015; Saide et al., 2015; Wang et al., 2018). Under polluted environments, BC has significant influence on pollution development (Ding et al., 2016; Peng et al., 2016; Lou et al., 2019). In addition, BC leads to visibility impairment because of its strong absorption of visible light (Watson, 2002) and has adverse impacts on human health due to its adsorption captivity (Janssen et al., 2011; Colicino et al., 2017). Nevertheless, due to lacking observational constraints and uncertainties in emission inventories, large uncertainties still exist in BC emissions including absolute fluxes and relative source contributions of fossil versus biomass combustion, which will complicate our knowledge on the multiple BC effects. It is also necessary to clarify the contributions of different sources to BC in order to determine efficient emission mitigation strategies. At global scale, BC emission sources can be attributed to fossil fuels (~ 40 %), open biomass burning (~ 40 %) and biofuels (~ 20 %) (Ramanathan and Carmichael, 2008). However, these fractions vary significantly because of the substantial spatial and temporal variations in BC emissions (Venkataraman et al., 2005; Rehman et al., 2011; Cheng et al., 2013; Andersson et al., 2015).

Several source apportionments for quantitatively differentiating between biomass and fossil sources of ambient BC aerosol have been conducted using observation-based methods, such as isotope (e.g., radiocarbon) analysis technique and light-absorbing property analysis (Sandradewi et al., 2008; Gustafsson et al., 2009; Liu et al., 2014; Vaishya et al., 2017; Helin et al., 2018; Kalogridis et al., 2018; Mousavi et al., 2018; Jing et al., 2019; Kant et al., 2020). For example, Andersson et al. (2015) presented dual carbon isotope constrained BC source apportionment in three key hotspot regions in China during a severe haze event, finding that biomass burning contributed ~ 30 % to BC whereas fossil fuel sources were dramatically different between north and south. Aethalometer model was adopted to analyze light absorption at multi-wavelengths to assess the fossil fuel and biomass combustions contributions to BC in Delhi, revealing that the contribution of biomass burning was 28 % in average (Dumka et al., 2018). Mousavi et al. (2019) apportioned BC in the Milan metropolitan area to the fossil fuel and biomass burning emission using the Aethalometer model with the absorption Ångström exponent (AAE) values derived from the [14]C radiocarbon analysis, highlighting the significant impact of

residential wood burning on BC. Such observation-based source apportionment methods are powerful to understand the BC sources at given receptor locations. However, the methods are highly dependent on accurate observations with high temporal resolution, which is unavailable for most regions without the measurement instruments. For example, the isotope method, especially the $^{14}$C analysis, is costly and lack of high temporal resolution. Therefore, although many observation-based BC source apportionments have been carried out, the source-based method can still be a strong supplement. For example, Winiger et al. (2019) conducted observation-based source apportionment of circum-Arctic BC with carbon isotope analysis and found that comparison of a Lagrangian atmospheric transport model (FLEXPART-ECLIPSE-GFED) predictions with the observations agreed well with each other for BC concentrations, with larger discrepancies for (fossil/biomass burning) sources, indicating the misallocations of emissions in the emission inventories.

Source-oriented modeling, which estimates pollution levels and identifies sources using chemical transport models (CTMs) with the inputs of emission inventory and meteorology, is another useful tool to study potential factors deriving BC. Such source apportionment technique has been developed and used for direct source apportionment of PM in more than a decade (Kleeman et al., 2007; Ying et al., 2008; Zhang et al., 2014). For example, Hu et al. (2015) found that residential emission was the major contributor to BC in spring and winter while industrial emission was important in summer and fall in China. Guo et al. (2017) quantified the contributions of different sources in North India and found that industry was the largest source for BC. Although the source-oriented modeling is powerful, limitations exist such as the inability to take into account unknown sources and the imprecise information on emission inventories and meteorology. The method is highly dependent on accuracy of emission inventory, which is unfortunately an enormous challenge. A pollution source not in the emission inventory will not emerge as a contributor to the CTM results. Taking into account the advantages and disadvantages of the observation- and modeling-based methods, a combination of the two methods can be a complement to each other for providing reliable and reasonable information on pollution sources and contributions.

China is the largest source of BC aerosols in the world (Wang et al., 2012; Bond et al., 2013; Huang et al., 2016), remarkable influences of BC on air quality, weather condition and climate change were revealed in China (Menon et al., 2002; Ding et al., 2016; Huang et al., 2016; Yang et al., 2017). Spatiotemporal distributions and regional transport mechanisms of BC in China as well as their affecting factors have been widely investigated with field measurements or model simulations (Cao et al., 2010; Wu et al., 2013; Wang et al., 2015; Zhang et al., 2019; Zheng et al., 2019; Deng et al., 2020). In contrast, source apportionment studies on BC aerosols in China are still limited and mostly distributed in heavily polluted areas (Chen et al., 2013; Andersson et al., 2015; Li, K. et al., 2016; Li, N. et al., 2016; Yu et al., 2018; Jing et al., 2019). In this study, the observation-based method was combined with the source-oriented modeling to quantify the contributions of different sources to BC in a relatively clean region in China. The results of the two source apportionment methods were inter-compared. Temporal variability, potential sources and transport pathways of BC from fossil fuel and biomass burning were also characterized. The findings help better understand main sources and relative contributions of BC and provide valuable

information to adopt effective emission reduction measures to control BC pollution in not heavily polluted regions.

## 2 Methodologies

### 2.1 Observation site and measurements

The field campaign was performed in Institute of Urban Environment, Chinese Academy of Sciences (118°03'E, 24°36'N) in the coastal city Xiamen in China (Deng et al., 2016, 2020). Xiamen is located in the Western Taiwan Strait region, which is adjacent to the Yangtze River Delta region (YRD) and the Pearl River Delta region (PRD) (Fig. 1). Xiamen has small local emission of BC (Fig. 1) and better diffusion condition compared to some developed cities in East China, which might lead to a lower BC concentration in Xiamen (Deng et al., 2020). However, Xiamen is often affected by emissions from polluted areas by long-range transport under the influence of East Asia monsoon (Deng et al., 2020). Therefore, conducting source apportionment of BC over Xiamen is very representative to improve our understanding on the sources of BC and their transport characteristics in relatively clean region.

The observation site (10 m above sea level) lies approximately 15 km away from the downtown to the southeast. None of large industrial sources was within 10 km away and there were only a few construction and traffic sources. The measurement instruments were arranged on the rooftop (8 m above ground level) of the site. Real-time measurements of BC mass concentration were conducted with a seven-wavelength (370, 470, 520, 590, 660, 880 and 950 nm) aethalometer (AE31, Magee Scientific) in January–December 2014. Aethalometer with a $PM_{2.5}$ cut-off inlet worked at a flow rate of 5 L $min^{-1}$ and estimated light attenuation under the principle of optical transmission (Hansen et al., 1984). BC concentration was then calculated according to the light attenuation. The concentration measured at 880 nm is considered as the standard value of atmospheric BC because BC is the predominant light-absorbing species at this wavelength with little impact from other compounds (Ganguly et al., 2005). The method reported in Virkkula et al. (2007) was applied to correct BC mass concentration due to shadowing effects and multiple scattering effects.

### 2.2 Observation-based source apportionment

Observation-based source apportionment of BC in Xiamen was performed with the Aethalometer method. The method based on the two-component assumption has been widely adopted to assess the contribution from fossil fuel combustion and biomass burning (Sandradewi et al., 2008; Favez et al., 2010; Liu et al., 2014; Rajesh and Ramachandran, 2017; Martinsson et al., 2017; Dumka et al., 2018; Helin et al., 2018; Mousavi et al., 2019; Mbengue et al., 2020). The Aethalometer method apportions the total BC to $BC_{ff}$ (BC emitted by fossil fuels) and $BC_{bb}$ (BC emitted by biomass burning) contributions. $BC_{ff}$ and $BC_{bb}$ are expressed as follows:

$$BC_{ff} = BC \times \frac{b_{abs,ff}(\lambda)}{b_{abs}(\lambda)} \tag{1}$$

$$BC_{bb} = BC \times \frac{b_{abs,bb}(\lambda)}{b_{abs}(\lambda)} \tag{2}$$

where $b_{abs}(\lambda)$ is light absorption at wavelength of $\lambda$, $b_{abs,ff}$ and $b_{abs,bb}$ are light absorption coefficients for fossil fuel and biomass burning, respectively. The light absorption is depending on the wavelength, satisfying the following relation:

$$\frac{b_{abs}(\lambda_1)}{b_{abs}(\lambda_2)} = \left(\frac{\lambda_1}{\lambda_2}\right)^{-\alpha} \tag{3}$$

where $\alpha$ is the AAE value. $b_{abs}$ is assumed to apportion to $b_{abs,ff}$ and $b_{abs,bb}$ contributions in the two-component method (Sandradewi et al., 2008):

$$b_{abs}(\lambda) = b_{abs,ff}(\lambda) + b_{abs,bb}(\lambda) \tag{4}$$

Using Eqs. (3)–(4) and $b_{abs}$ the measured at two different wavelengths, the fossil fuel and biomass burning contribution can be derived using the following equations:

$$\frac{b_{abs,ff}(\lambda_1)}{b_{abs,ff}(\lambda_2)} = \left(\frac{\lambda_1}{\lambda_2}\right)^{-\alpha_{ff}} \tag{5}$$

$$\frac{b_{abs,bb}(\lambda_1)}{b_{abs,bb}(\lambda_2)} = \left(\frac{\lambda_1}{\lambda_2}\right)^{-\alpha_{bb}} \tag{6}$$

$$b_{abs,bb} = \frac{b_{abs}(\lambda_1) - b_{abs}(\lambda_2) \cdot \left(\frac{\lambda_1}{\lambda_2}\right)^{-\alpha_{ff}}}{\left(\frac{\lambda_1}{\lambda_2}\right)^{-\alpha_{bb}} - \left(\frac{\lambda_1}{\lambda_2}\right)^{-\alpha_{ff}}} \tag{7}$$

$$b_{abs,ff} = \frac{b_{abs}(\lambda_1) - b_{abs}(\lambda_2) \cdot \left(\frac{\lambda_1}{\lambda_2}\right)^{-\alpha_{bb}}}{\left(\frac{\lambda_1}{\lambda_2}\right)^{-\alpha_{ff}} - \left(\frac{\lambda_1}{\lambda_2}\right)^{-\alpha_{bb}}} \tag{8}$$

where $\alpha_{ff}$ and $\alpha_{bb}$ are the AAE values for fossil fuel and biomass burning, respectively. For preselected $\alpha_{ff}$ and $\alpha_{bb}$ values, $b_{abs,ff}$ and $b_{abs,bb}$ can be calculated by Eqs. (7)–(8). $BC_{ff}$ and $BC_{bb}$ can be obtained by combining all above equations and assumed values for $\alpha_{ff}$ and $\alpha_{bb}$. In this study, 470 nm and 950 nm were selected as $\lambda_1$ and $\lambda_2$ in accordance with previous studies (Sandradewi et al., 2008; Favez et al., 2010; Zotter et al., 2017; Helin et al., 2018; Kalogridis et al., 2018).

In the Aethalometer model, one of the largest uncertainties is related to the choice of the $\alpha_{ff}$ and $\alpha_{bb}$ values (Sciare et al., 2011; Healy et al., 2017; Zotter et al., 2017; Helin et al., 2018). The site-specific $\alpha$ values are affected by the type of fuel, combustion regime, and the mixing state of BC aerosols with non-absorbing materials (Favez et al., 2010; Lack and Langridge, 2013; Garg et al., 2016). All of these factors increase the uncertainty of observation-based source apportionment.

In the literature, fixed $\alpha_{ff}$ and $\alpha_{bb}$ values were commonly used in source apportionment studies for simplicity (Favez et al.,

2010; Herich et al., 2011; Sciare et al., 2011; Harrison et al., 2013; Fuller et al., 2014; Rajesh and Ramachandran, 2017; Zotter et al., 2017; Helin et al., 2018). For example, Sandradewi et al. (2008) suggested that $\alpha_{ff}$ was 1.1 and $\alpha_{bb}$ was 1.8–1.9 from the light absorption at 470 and 950 nm. Zotter et al. (2017) suggested the AAE values in the Aethalometer model was site and source specific and recommended using $\alpha_{ff}$ of 0.9 and $\alpha_{bb}$ of 1.68 in Switzerland. Based on a comprehensive investigation on previous studies using the Aethalometer model (Table S1), $\alpha_{ff}$ and $\alpha_{bb}$ values were most commonly in the

range of 0.9–1.1 and 1.7–2.2, respectively. In this work, the empirical value of $\alpha_{ff}$ and $\alpha_{bb}$ is adopted as 1.0 and 2.0 following many previous studies (Kirchstetter et al., 2004; Favez et al., 2010; Crippa et al., 2013; Fuller et al., 2014; Crilley et al., 2015; Petit et al., 2017; Vaishya et al., 2017; Xiao et al., 2020). In addition, in order to test the impact of AAE values on performance of the Aethalometer model, a sensitivity analysis with various combinations of AAE pairs was implemented. Except for the base assessment with $\alpha_{ff} = 1.0$ and $\alpha_{bb} = 2.0$, the analysis was conducted by changing the $\alpha_{ff}$ from 0.9 to 1.1

and the $\alpha_{bb}$ from 1.7 to 2.2.

## 2.3 Potential sources of BC$_{ff}$ and BC$_{bb}$

The conditional probability function (CPF) was used to investigate the possible predominant directions of local sources of BC$_{ff}$ and BC$_{bb}$ relative to wind directions in different seasons (Ashbaugh et al., 1985). The CPF is calculated as:

$$CPF_{\Delta\theta} = m_{\Delta\theta} / n_{\Delta\theta} \tag{5}$$

where $n_{\Delta\theta}$ is the total occurrences from wind sector $\Delta\theta$ and $m_{\Delta\theta}$ is occurrences from the same wind sector with the BC$_{ff}$ (BC$_{bb}$) concentration exceeding the threshold criterion. The CPF analysis was also performed for the ratio of BC$_{bb}$ to BC (BC$_{bb}$/BC) to analyze the impact of local sources on the contribution from biomass burning. In this analysis, a threshold criterion of the top 25 % concentration (ratio) was chosen (Deng et al., 2020).

Backward trajectories were simulated with the Hybrid Single Particle Lagrangian Integrated Trajectory model (HYSPLIT)

from NOAA/ARL to characterize the regional sources and transport of air masses arriving in Xiamen (Stein et al., 2015). Five-day backward trajectories ended at the height of 500 m were calculated every hour using the Global Data Assimilation System (GDAS) reanalysis meteorological dataset with a $1° \times 1°$ latitude-longitude resolution. Hourly trajectory endpoints implying the geographical distribution and the height of the air parcel were derived from the model. Trajectory clusters were then obtained from cluster analysis, which was performed based on the inputs of hourly backward trajectories with the

TrajStat plugin of the Meteoinfo (http://www.meteothink.org/) software. Four clusters were obtained for each season with the clustering option of angle Euclidean distance. The outflow regimes for air masses to the receptor site with the potential origins were traced with the trajectory clusters.

Potential regional source contributions of BC$_{ff}$ and BC$_{bb}$ were further identified with the potential source contribution

function (PSCF) method on basis of the backward trajectories. PSCF is a widely adopted tool to identify regional source distributions of air pollutants at a receptor site (Hopke et al., 1995; Bari et al., 2015; Zhang et al., 2017). The study domain is divided into $i \times j$ grid cells and PSCF values can be calculated as follows:

$$PSCF_{i,j} = m_{i,j} / n_{i,j} \tag{6}$$

where $n_{i,j}$ is the number of endpoints that in the $ij$th grid cell and $m_{i,j}$ is the number of endpoints for the same grid cell that have $BC_{ff}$ ($BC_{bb}$) concentration higher than a criterion. These grid cells with high PSCF values are the maximum probability potential source areas contributing to high $BC_{ff}$ ($BC_{bb}$) mass concentrations at the receptor location. In this work, the top 25 % concentrations were set as the threshold. The study domain covered 10°–55° N and 80°–140° E, which composes 10800 grid cells with the size of 0.5° × 0.5° latitude and longitude.

To minimize the uncertainty in grid cells with low $n_{i,j}$, an empirical weight function $w_{i,j}$ was multiplied with the PSCF values. $w_{i,j}$ was defined as follows:

$$w_{i,j} = \begin{cases} 1.00 & n_{i,j} > 3n_{ave} \\ 0.70 & 1.5n_{ave} < n_{i,j} \leq 3n_{ave} \\ 0.42 & n_{ave} < n_{i,j} \leq 1.5n_{ave} \\ 0.05 & n_{i,j} \leq n_{ave} \end{cases} \tag{7}$$

It is difficult for PSCF method to identify the source intensity and separate strong sources and weak sources. Therefore, concentration-weighted trajectory (CWT) model was also performed to overcome this limitation. In this method, each grid cell is assigned a weighted concentration by averaging the sample concentrations that have associated trajectories crossing the grid cell (Hsu et al., 2003). The average weighted concentration $C_{i,j}$ in the $ij$th grid cell was calculated as follows:

$$C_{i,j} = \frac{1}{\sum_{l=1}^{M} \tau_{i,j,l}} \sum_{l=1}^{M} C_l \tau_{i,j,l} \tag{8}$$

where $M$ is the total number of trajectories, $C_l$ is the observed $BC_{ff}$ ($BC_{bb}$) concentration at receptor site on arrival of trajectory $l$ and $\tau_{i,j,l}$ is the number of endpoints in the $ij$th grid cell of trajectory $l$. In general, the grid cells with high CWT values are high strength sources. The weighting function $w_{i,j}$ was also adopted in the CWT analysis to reduce the effect of the small values of $n_{i,j}$.

**2.4 Source-oriented modeling**

In this analysis, source apportionment of BC over Xiamen using an updated source-oriented Community Multiscale Air Quality Modeling System (CMAQ) model for primary particulate matter (CMAQ-PPM) (Hu et al., 2015; Guo et al., 2017) was also implemented in addition to the observation-based source apportionment. The CMAQ-PPM model was updated on basis of CMAQ v5.0.1, which was developed by the U.S. EPA Atmospheric Science Modeling Division. The photochemical

mechanism and aerosol chemistry mechanism adopted in this study were SAPRC-11 and AERO6, respectively. In the source-oriented model, tagged non-reactive PM tracers are used to estimate the source contributions of PPM and its chemical components. The PM tracers are set to undergo the same atmospheric processes as other species. The emissions of the tracers are set to 0.001 % of the PPM emissions from each corresponding source sector and region. It ensures that the tracers will not significantly change the particle mass and size. After scaling up by $10^5$, the simulated tracer concentration represents the PPM concentrations from a specific source type/region. The concentrations of the inert chemical components in PPM can be estimated with source specific emission profiles as follows:

$$C_{i,j} = A_{i,j} \times PPM_i \qquad (9)$$

where $C_{i,j}$ is the concentration of the $j$th component from the $i$th source, $A_{i,j}$ is the ratio of the $j$th component in PPM mass from the $i$th source and $PPM_i$ is the simulated concentration for the $i$th source. Detailed descriptions of the model can be found in Hu et al. (2015).

The sourced-oriented modeling with tagged tracers is similar to the particulate source apportionment technology (PSAT). However, PSAT does not track the species from different sources directly in each time step. Instead, it allocates the changes of bulk concentrations to different sources after each time step based on the ratio of each source to total emissions. Comparing with the source-oriented model, the brute force method (BFM) is more suitable to estimate the change of PM due to proposed emission control measures than to determine the contributions of certain sources because removal of PM emissions could affect the transport, chemistry, deposition and interactions with meteorology although they are not chemically reactive (Zhang and Ying, 2011). The results simulated with the BFM is different from "source apportionment" since the summation of the contributions of all source categories will not always equal the total concentration. In addition, the BFM needs to repeat chemical transport model simulations multiple times and greatly increases the computational cost.

Regional distributions of BC from different categories (sectors) as well as the source category (sector) contributions to BC at the receptor site were determined with the source-oriented CMAQ-PPM model. A 36 km horizontal resolution domain that covers China and surrounding countries in East Asia (Fig. 1) was applied. There are 18 vertical layers with surface layer thickness of 35 m and the overall model height of 20 km. The Weather Research & Forecasting model (WRF) v3.9.1 was utilized to generate meteorology inputs with initial and lateral boundary conditions from NCEP FNL reanalysis data from NCAR, which is available on $1° \times 1°$ grids continuously for every 6 h (http://dss.ucar.edu/datasets/ds083.2/). There are 29 vertical layers in the WRF domain. The first eight layers of the WRF and CMAQ domains are identical. The outputs of WRF were post-processed by Meteorology-Chemistry Interface Processor (MCIP) v4.2 to the format CMAQ requires. Anthropogenic emissions in China were generated according to the Multi-resolution Emission Inventory for China (MEIC) developed by Tsinghua University (http://www.meicmodel.org). Emissions from other countries and regions outside China were generated with the Regional emission inventory in Asia version 2 (REAS2) (Kurokawa et al., 2013). The fire emissions

were derived from the Fire Inventory from NCAR (FINN) based on satellite observations (Wiedinmyer et al., 2011). Anthropogenic emissions were grouped into four sectors including industrial, power, transportation and residential. Open burning emissions are considered as the fifth emission sector. Open biomass burning generally refers to open combustion of various biomass materials such as forest vegetation, crop residue and municipal solid waste (Permadi and Oanh, 2013). Sources from five sectors were further classified into three categories such as solid fossil fuel (i.e. coal) combustion, liquid fossil fuel combustion and biomass burning on basis of the energy consumption data provided by Wang et al. (2012). The performance of the source-oriented model on BC was evaluated by all available observations within China in Hu et al. (2015), which found that the model could well reproduce the BC concentrations and there was a good agreement between the BC observation and simulation. Spatial distribution of BC concentration over China are depicted in Fig. S1, suggesting that BC concentration in the Western Taiwan Strait region was much lower than that in other urban agglomerations in north China, east-central China and Sichuan Basin. However, Xiamen was with relatively higher abundance compared to surrounding areas.

## 3 Results and discussion

### 3.1 Light absorption-based source apportionment of BC

Fig. 2 demonstrates the temporal variations in daily mean concentrations of $BC_{ff}$ and $BC_{bb}$ with the $BC_{bb}$/BC fraction in Xiamen during the field campaign. The missing data were due to the instrument maintenance. Daily concentrations of $BC_{ff}$ and $BC_{bb}$ were $445 - 9545$ ng m$^{-3}$ (average: $2932 \pm 1444$ ng m$^{-3}$) and $334–4031$ ng m$^{-3}$ ($1340 \pm 542$ ng m$^{-3}$), respectively. Daily contribution of $BC_{bb}$ to total BC varied significantly in the range of $18.4 \% – 58.3 \%$, and daily $BC_{ff}$/BC fraction range from $41.7 \%$ to $81.6\%$. The annual average contribution of $BC_{bb}$ to BC was $33.3 \%$, much smaller than that of $BC_{ff}$ ($66.7 \%$), indicating the predominant contribution of fossil fuel combustion in Xiamen. The sensitivity of the Aethalometer model was investigated by using different $\alpha_{ff}$ and $\alpha_{bb}$ combinations (Fig. 3). The $BC_{ff}$/BC fraction increased with an increase in $\alpha_{ff}$ value, and there was even a more rapid increase in that fraction when $\alpha_{bb}$ increased. Contrarily, the $BC_{bb}$/BC fraction decreased with the increasing $\alpha_{ff}$ and $\alpha_{bb}$ values. In the sensitivity tests, the $BC_{ff}$ contributions were in the range from $42 \%$ ($\alpha_{ff} = 0.9$, $\alpha_{bb} = 1.7$) to $79 \%$ ($\alpha_{ff} = 1.1$, $\alpha_{bb} = 2.2$). The sensitivity analysis also indicates that the apportionment results are more sensitive to the changing $\alpha_{bb}$ values than to $\alpha_{ff}$ values. For example, keeping the $\alpha_{ff}$ value at 1.0, the $BC_{ff}$/BC increased from $46 \%$ to $75\%$ for $\alpha_{bb}$ value from 1.7 to 2.2. However, a slower increase in the $BC_{ff}$/BC fraction from $64 \%$ to $72 \%$ was found when $\alpha_{ff}$ value change from 0.9 to 1.1 by fixing the $\alpha_{bb}$ value at 2.0. It is different from previous studies over Granada in Spain (Titos et al., 2017) and Delhi in India (Dumka et al., 2018), which both found the Aethalometer model was more sensitive to $\alpha_{bb}$ than to $\alpha_{ff}$.

Source apportionment results under different levels of air pollutants (i.e. BC and PM$_{2.5}$) in each season were further

investigated to understand BC sources on pollution and clean days (Fig. 4). The data of $PM_{2.5}$ concentration measured by a Tapered Element Oscillating Microbalance (TEOM) sampler (RP1400, Thermo Fisher Scientific) were from Xiamen Environmental Monitoring Central Station. High-pollutant periods are the days with daily average concentration higher than the seasonal average plus one standard deviation, while low-pollutant periods are the days with daily average concentration lower than the seasonal average minus one standard deviation. Generally, source contributions of BC show obvious variability among different pollution levels in all seasons, and the $BC_{bb}$ percentage decreases with the increasing concentrations of BC and $PM_{2.5}$. Biomass burning contributed more during low-BC (30.8 % – 43.1 %) and low-$PM_{2.5}$ days (31.5 % – 40.7 %) compared to high-BC (24.8 % – 34.4 %) and high-$PM_{2.5}$ episodes (26.6 % – 36.1 %), implying that emissions from coal combustion and vehicle exhausts are major contributors of particulate pollution in Xiamen. The fractional contribution of fossil fuel to BC in Xiamen derived by the Aethalometer method in this work suggests a slighter larger role of fossil fuel compared to that (61 %) estimated according to the "bottom-up" emission inventories (Chen et al., 2013). However, it was similar to the contribution (~ 70 %) in YRD and PRD, which was estimated based on dual carbon isotope constrained source apportionment (Andersson et al., 2015). $BC_{ff}$ and $BC_{bb}$ percentages in different regions calculated with the Aethalometer method were summarized in Table S2 for comparison. $BC_{bb}$ fraction in Xiamen were overall larger than that in Nanjing in China and other sites in India, suggesting that contribution of biomass burning increases over the relatively clean region due to the weak emissions of traffic and coal combustion.

Fig. 5 illustrates the diurnal and monthly cycles of $BC_{ff}$ and $BC_{bb}$ concentrations as well as the relative contribution of biomass burning ($BC_{bb}$/BC) during the measurements. $BC_{ff}$ exhibited a pronounced diurnal variation, increasing steadily before dawn with the major morning peak (4427 ng m$^{-3}$) observed around 6:00 in the morning. The high $BC_{ff}$ concentrations at the observation site from late night to the early morning (~ 21:00 to 8:00) may ascribe to enhanced traffic emissions from diesel trucks during nighttime and cars during rush hours. The heavy diesel trucks, which are major emission sources of $BC_{ff}$, were allowed to enter the city from 22:00 to 7:00. Therefore, $BC_{ff}$ decreased during daytime and reached the diurnal minimum of 1950 ng m$^{-3}$ at 13:00 in the afternoon. $BC_{bb}$ exhibited a diurnal trend that was different with $BC_{ff}$. The morning peak (1755 ng m$^{-3}$) at 6:00 was also found for $BC_{bb}$. However, $BC_{bb}$ concentration kept a steady state rather than increase after 20:00 since $BC_{bb}$ was not influenced by traffic-related emission. Clear diurnal variation in contribution of $BC_{bb}$ to total BC was found. The $BC_{bb}$ fraction reached its valley of 30.4 % at 8:00, increased due to the decrease in traffic emission and maximized with the ratio of 37.9 % at 19:00 in the evening due to increases in biomass burning activities. The diurnal cycles of $BC_{ff}$ and $BC_{bb}$ were affected by not only the BC emission trend but also the evolution of atmospheric boundary layer. According to our previous study on atmospheric boundary layer height in Xiamen (Deng et al., 2020), the boundary layer height was ~three times larger in the afternoon than that in the early morning, leading to the better diffusion conditions in the afternoon.

Monthly mean $BC_{bb}$ concentration peaked with value of 1979 ng m$^{-3}$ in December and reached its valley of 923 ng m$^{-3}$ in

June. The monthly pattern of $BC_{ff}$ was similar but a bit different with that of $BC_{bb}$. The maximum monthly mean $BC_{ff}$ concentration was 3636 ng m$^{-3}$ in March, while the minimum was 1881 ng m$^{-3}$ in February. The valley of $BC_{ff}$ concentration occurring in February was maybe because of the lack of vehicle (e.g., diesel trucks) emissions around the Spring Festival holiday, which again prove the conjecture in the diurnal pattern of $BC_{ff}$. Similar to the seasonal pattern of absorption Ångstrom exponent (Qiu et al., 2019), noticeable seasonal variation in the $BC_{bb}$/BC fraction was found. Winter (December–February) had the largest $BC_{bb}$ contribution (39.9 %), followed by fall (September–November) (32.1 %), spring (March–May) (31.1 %) and summer (June–August) (29.6 %). The much larger contribution in winter are possibly due to the enhanced source from open-field biomass and domestic burning in China (He et al., 2011). The higher $BC_{bb}$ concentration and contribution lasted from fall to early winter, consistent with previous emission inventory of biomass burning, which found higher BC emissions from November to February than other months (He et al., 2011). Unlike $BC_{ff}$, $BC_{bb}$ exhibited an increase trend in July, leading to a relatively large contribution of $BC_{bb}$. It might be affected by the long-range transport of air pollutants emitted from biomass burning in Southeast Asia under the control of summer monsoon (Qiu et al., 2019). The monthly variation in boundary layer height, which was larger in warm season and smaller in cold season, also affected the monthly patterns of $BC_{ff}$ and $BC_{bb}$ (Deng et al., 2020).

### 3.2 Sources and transport pathways of $BC_{ff}$ and $BC_{bb}$

The CPF results for the top 25 % thresholds of concentrations of $BC_{ff}$ (3797 ng m$^{-3}$) and $BC_{bb}$ (1813 ng m$^{-3}$) as well as $BC_{bb}$ contribution (45 %) over different periods are shown in Fig. 6. In the whole year, high $BC_{ff}$ concentrations were mainly associated with winds from west-southwest to north-northeast with wind speed (ws) < 2 m s$^{-1}$ (Fig. 6a). Particularly, high $BC_{ff}$ concentrations were most remarkably distributed in winds from the northwest at low ws (< ~1 m s$^{-1}$) and to a lesser extent from the west and north-northeast at moderate ws (< 3 m s$^{-1}$). It implies the impacts of local sources such as the traffic emissions to the northwest of the site within a short distance. The CPF pattern for $BC_{bb}$ was similar to but not the same with that of $BC_{ff}$ (Fig. 6b). In addition to northwesterly wind with low ws, northeasterly and easterly winds with ws < 5 m s$^{-1}$ were also accompanied by high $BC_{bb}$ concentration. Correspondingly, the CPF plot for $BC_{bb}$/BC fraction implies the significant influence of east-northeasterly wind with ws > 2 m s$^{-1}$ on large contribution of biomass burning (Fig. 6c). In addition, northerly wind with wind speed > ~4 m s$^{-1}$ were also frequently associated with large $BC_{bb}$ fraction. CPF patterns presented obvious seasonality. For $BC_{ff}$ and $BC_{bb}$, the CPF distributions over spring, summer and fall were similar and the high concentrations were mainly associated with northwesterly wind with ws < 2 m s$^{-1}$. However, in winter, additionally with wind from northwest, high $BC_{ff}$ and $BC_{bb}$ concentrations were also frequently associated with wind from southwest and west with ws < 3 m s$^{-1}$. For $BC_{bb}$/BC fraction, large fractions were mainly distributed in northeasterly wind with ws > 4 m s$^{-1}$ and most remarkably distributed in winds from the northeast at high ws (> 6 m s$^{-1}$) in spring and summer. In fall,

east-northeasterly wind with ws > 3 m s$^{-1}$ was more frequently associated with large contribution of biomass burning. However, in winter, northerly wind with ws > 4 m s$^{-1}$ and northeasterly wind with ws > 2 m s$^{-1}$ were most remarkably associated with large BC$_{bb}$ percentage.

Seasonal clusters of backward trajectories obtained by the HYSPLIT model with the average BC$_{bb}$ contributions are illustrated in Fig. 7. Mean concentrations of BC$_{ff}$, BC$_{bb}$ and BC of each cluster in different seasons are summarized in Table S3. It is clearly shown that originations and transport pathways of air masses arriving in Xiamen exhibited distinct seasonal variations. In summer, air masses were characterized by a predominance of southerly origination. In contrast, in other seasons, air masses from the north had a dominant position, which was particularly the case in winter. Generally, air masses from the northern inland region such as north China and east-central China had larger biomass burning contributions comparing to those from the seas such as East China Sea and South China Sea, since there are dense emissions of biomass burning in northern and eastern China including Hebei, Henan, Shandong and Jiangsu (Huang et al., 2012; Wu et al., 2018). In spring, the eastern cluster (C4) originating from the East China Sea had the lowest BC$_{bb}$ fraction (31 %). However, the northern cluster (C3) originating from Siberia and passing through Mongolia and north and east China had much larger biomass burning contribution (42 %) in comparison to the other clusters. In summer, northeastern coastal cluster (C4) originating from the East China Sea and passing along with East China Coast region had larger biomass burning contribution (38 %). The north cluster passing through Jiangsu, Zhejiang and north Fujian province also had relatively higher BC$_{bb}$ fraction (35 %). In fall, the northern inland cluster (C3) originating from Siberia and passing through the heavily polluted areas such as North China Plain and YRD were associated the largest biomass burning contribution (40 %), followed by the other long-range inland cluster (C2) with the BC$_{bb}$ fraction of 36 %. In winter, similar to spring and fall, the northern cluster from Siberia (C2) had the largest biomass burning contribution (48 %). Contrarily, the northeastern marine air masses passing along with the coastal region had the lowest BC$_{bb}$ fraction (35 %).

Potential sources of BC$_{ff}$ and BC$_{bb}$ in Xiamen with their contributions were characterized with CWT and PSCF analyses and the results are presented in Fig. 8 and Fig. S2. According to the PSCF and CWT maps of BC$_{ff}$ (Fig. S2a and Fig. 8a), the strong potential sources probabilities for BC$_{ff}$ distributed to southwest of Xiamen, including southwest Fujian province as well as Guangdong province. Significant potential sources were also located in Hubei, Anhui, Jiangxi and Henan provinces in east-central China and Hebei and Shandong provinces in north China, again implying the influences of long-range transport on BC$_{ff}$ in Xiamen. Southeast Asia with strong regional BC emissions (Permadi et al., 2018) was also indicated as the potential source regions. For BC$_{bb}$, similar to BC$_{ff}$, the PSCF and CWT distributions (Fig. S2b and Fig. 8b) show that the exogenous potential sources were mainly distributed in east-central China, which belonged to the major areas of biomass burning in China (Yan et al., 2006; Huang et al., 2012; Wu et al., 2018). Guangdong province in south China was also suggested as the source of BC$_{bb}$. Unlike BC$_{ff}$, the strong potential sources probabilities from Southeast Asia to BC$_{bb}$ were less significant.

Fig. 9 and Fig. S3 depict the seasonal CWT and PSCF distributions for $BC_{ff}$ and $BC_{bb}$. Source distributions of both $BC_{ff}$ and

$BC_{bb}$ in different seasons significantly varied due to the variability in the airflows. In spring, the terrestrial contributions

from Guangdong province and north China to $BC_{ff}$ and $BC_{bb}$ were significant. In addition, high PSCF and CWT values for

summer $BC_{bb}$ were also found in north China. In fall season, similar to spring, high PSCF and CWT values for $BC_{ff}$ and

$BC_{bb}$ were distributed in east-central China. In winter, the main potential sources for $BC_{ff}$ and $BC_{bb}$ were also located in

east-central China. The potential sources for $BC_{bb}$ in central China were much stronger in winter than that in other seasons.

The East China Sea and South China Sea were also indicated as the potential source areas for $BC_{ff}$ and $BC_{bb}$ by the PSCF

and CWT analysis. However, they should not be real source areas and were identified due to the trailing effect (Lee et al.,

2014; Deng et al., 2020).

## 3.3 Source-oriented modeling-based source apportionment of BC

Relative source contributions to BC in Xiamen from different source sectors and fuel catalogs were assessed with the

source-oriented CMAQ-PPM model. Fig. 10 illustrates the seasonal and annual average contributions of each source sector

in Xiamen. Overall, transportation, residential and open burning sectors were the major sources of BC, with the annual

contributions of 45.3 %, 30.1 % and 17.6 %, respectively. By comparison, power plants and industrial sectors made minor

contributions to BC, accounting for 3.4 % and 3.6 %, respectively. The transportation sector was the dominant source in all

seasons, especially in summer, contributing 36.5 % – 56.6 % to total BC. The residential sector contributing 20.5 % – 37.2 %

was the second largest source in all seasons except spring. On contrary, power plants and industrial sectors were minor

sources in all seasons, with the seasonal contributions of 2.2 % – 6.2 % and 2.8 % – 4.6 %, respectively. Obvious seasonal

pattern of contribution of open burning was found. In spring and summer, open burning played a vital role by contributing

35.5 % and 17.8 %, respectively. However, its relative contributions dramatically decreased to 7.6 % in fall and 7.5 % in

winter. Source contributions of the five sectors to BC over China in different periods are depicted in Fig. S4. The remarkable

seasonal and spatial variations from open burning are consistent with those derived in previous study (Hu et al., 2015). In

spring, strong open burning in south China might significantly influence BC concentrations in the surrounding regions near

the sources, which would contribute to larger biomass burning contribution in Xiamen. Intensive open burning in South Asia

and Southeast Asia countries (Sharma et al., 2010; Vadrevu et al., 2015; Sharma et al., 2020) in spring also affected biomass

burning contribution in Xiamen through long-range transport (Fig. 9).

Seasonal variations in simulated relative contributions of three fuel catalogs (i.e., coal, liquid fossil fuel and biomass) to BC

in Xiamen were demonstrated in Fig. 11. For the entire year, liquid fossil fuel combustion had the largest contribution

(46.5 %), followed by biomass burning (32.6 %) and coal combustion (20.9 %). Contributions of different fuel catalogs

exhibited distinct seasonality. Seasonal contributions of coal combustion were in the order of winter (27.8 %) > fall

(23.4 %) > spring (15.6 %) > summer (14.5 %). For liquid fossil fuel combustion, its largest contribution (57.5 %) was in summer and its smallest contribution (37.4 %) was in spring. Seasonal contributions of biomass burning were in the range of 25.7 % – 47.0 %, and the average contribution was much larger in spring than in other seasons. Contribution of fossil fuel combustion, which is the sum of liquid fossil fuel and coal combustion, followed the order of winter > fall > summer > spring.

The simulated contributions were compared with the source apportionment results estimated according to light-absorption properties. For the whole year, the annual average relative contributions of $BC_{ff}$ and $BC_{bb}$ derived by the source-oriented model were 67.4% and 32.6%, respectively. They were very close to the results (i.e., 66.7% for $BC_{ff}$ and 33.3% for $BC_{bb}$) obtained by the Aethalometer method. The overall consistence of the two apportionment methods confirms that the source apportionment results in Xiamen from this study are reasonable and benefiting future emission-control strategies. Simulated

contributions of $BC_{ff}$ and $BC_{bb}$ were 72.0 % and 28.0 % in summer, and 73.9 % and 26.1 % in fall. The simulated contributions in summer and fall were comparable to those derived by the Aethalometer method, and the discrepancies between the results from the two methods were 1.7 % in summer and 6.0 % in fall. However, there were considerable differences between the BC source apportionment results from the Aethalometer method and source-oriented CMAQ model in winter and spring. Relative contributions of $BC_{ff}$ and $BC_{bb}$ derived by the source-oriented modeling were 74.3 % and

25.7 % in winter, and 47.0 % and 53.0 % in spring. Simulated $BC_{ff}$ ($BC_{bb}$) contribution was 14.3 % larger (smaller) in winter and 15.9 % smaller (larger) in spring compared to the observation-based results. Both the uncertainties in the Aethalometer method and the emission inventory used in the source-oriented model may lead to the gap between observation-based and model results. For example, due to lack of the auxiliary measurements, the AAE values of $BC_{ff}$ and $BC_{bb}$ adopted in this study were based on a comprehensive literature review. It would lead to uncertainties in the apportionment results from the

Aethalometer model, as discussed in Section 3.1. On the other hand, the large gap in spring and winter may partly be resulted from the uncertainties in satellite-based inventory of biomass-burning emissions in South Asia and China (Wiedinmyer et al., 2011; Huang et al., 2012; Zhou et al., 2016). In China, some open burning activities such as local/small-scale open burning and smoldering are important sources of biomass-burning BC, which was particularly the case in winter. However, these burning activities are difficult to be accurately detected by satellite, leading to considerable uncertainties in biomass-burning

emissions. In addition, low resolution of simulation could lead to uncertainties in model results. The peak values close to emission sources may not be captured after the dilution of emission in large grid cells. In future, high resolution is suggested as long as high-resolution emission inventories are available.

**4 Conclusions**

In this study, the observation-based light absorption and source-oriented modeling were combined to reveal the contributions

of biomass burning and fossil fuel combustion to ambient BC aerosol as well as their temporal variations in a relatively clean region in China. The annual average concentration of $BC_{ff}$ and $BC_{bb}$ identified by the Aethalometer method with $\alpha_{ff}$ = 1.0 and $\alpha_{bb}$ = 2.0 were 2932 ± 1444 ng m$^{-3}$ and 1340 ± 542 ng m$^{-3}$, accounting for 66.7 % and 33.3 % of total BC, respectively. A sensitivity analysis conducted by changing the $\alpha_{ff}$ and $\alpha_{bb}$ values suggested that increase in $\alpha_{ff}$ or $\alpha_{bb}$ values would lead to increase in $BC_{ff}$ against $BC_{bb}$, and the Aethalometer method was more sensitive on changes in $\alpha_{bb}$ rather than $\alpha_{ff}$. For biomass

burning contribution, its highest level occurred in the evening rush hour, while the maximum seasonal value was in winter. East-northeasterly and northerly wind was more likely to result in large biomass burning contribution. Air masses from the northern inland region including north China and east-central China had larger biomass burning contributions. Potential sources for $BC_{ff}$ and $BC_{bb}$ indicate the impact of long-range transport from north and east-central China and Southeast Asia. Based on the source-oriented model, the transportation, residential and open burning sectors were the larger contributors to

BC compared to the power and industrial sectors. The largest contribution of liquid fossil fuel combustion to BC was identified by the source-oriented model, followed by biomass burning and coal combustion. The simulated contributions of $BC_{ff}$ and $BC_{bb}$ were 67.4 % and 32.6 %, respectively, close to the results of the Aethalometer method. The simulated contributions in summer and fall were comparable to those derived by the Aethalometer method. However, the differences between the two apportionment methods in winter and spring were considerable. The discrepancies between the two source

apportionment methods suggest accurate emission inventory with higher spatiotemporal resolution is required in future studies. Source apportionment of BC in Xiamen from both light absorption observation and source-oriented modeling indicate that the fossil fuel sources should be strictly controlled to limit the BC pollution. The findings also suggests that it is essential to reduce biomass burning in future pollution management strategies.

*Data availability.* The data is available upon request from Junjun Deng (dengjunjun@tju.edu.cn).

*Author contribution.* JD and HZ designed the experiments and carried them out. JD and WZ performed the analysis of observation. HG and HZ performed the source-oriented modeling. JZ, WH, LW, XW and PF provided suggestions for data analysis. JD prepared the manuscript with contributions from all co-authors.


*Competing interests.* The authors declare that they have no conflict of interests.

*Acknowledgments.* This study was supported by National Key Research and Development Program of China (2019YFA0606801) and National Natural Science Foundation of China (21607148).

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

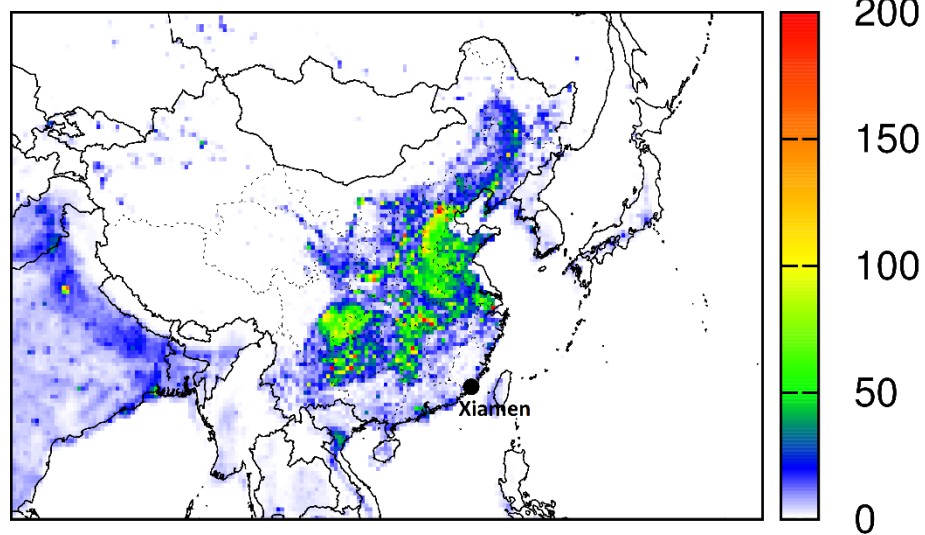

**Figure 1. Location of Xiamen, China, with spatial distribution of annual average BC emission rate (g s⁻¹). BC emission data in China are from the MEIC inventory developed by Tsinghua University.**

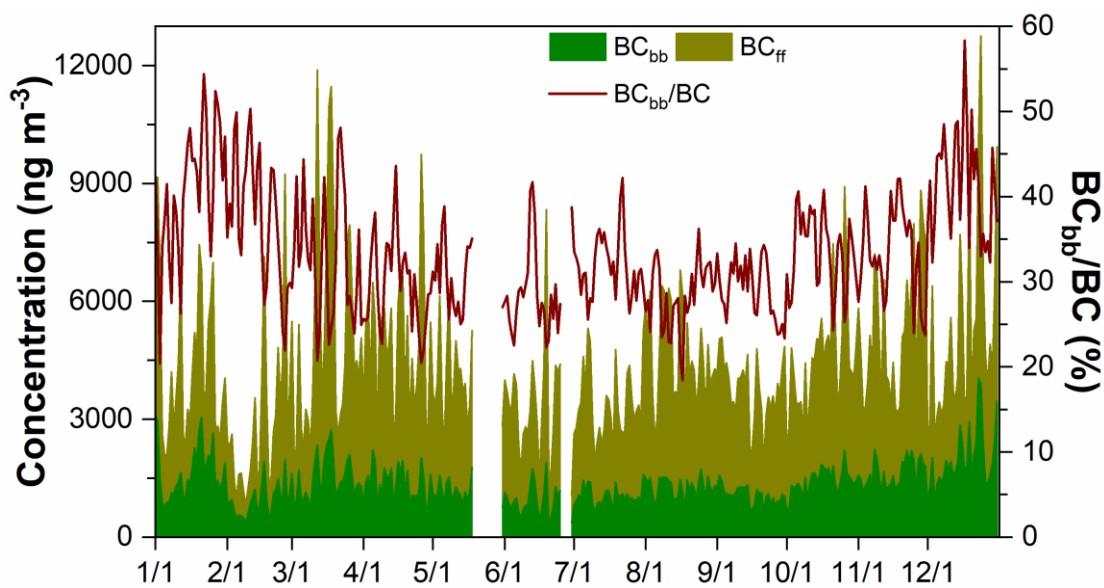

**Figure 2. Daily BC$_{ff}$ and BC$_{bb}$ concentrations and BC$_{ff}$/BC fraction in Xiamen in 2014.**

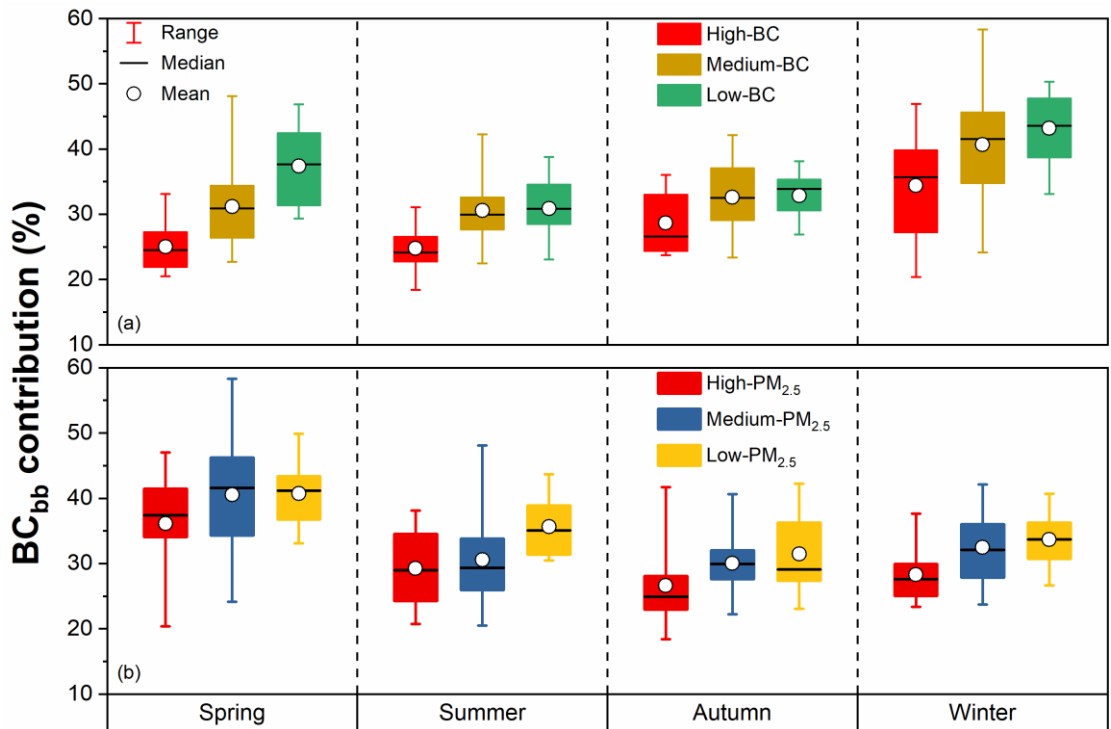

**Figure 3. Variations in the (a) BC$_{ff}$/BC and (b) BC$_{bb}$/BC fractions with $\alpha_{ff}$ and $\alpha_{bb}$.**

**Figure 4. Box plots of BC$_{bb}$ contribution with different concentrations of (a) BC and (b) PM$_{2.5}$ in different seasons.**

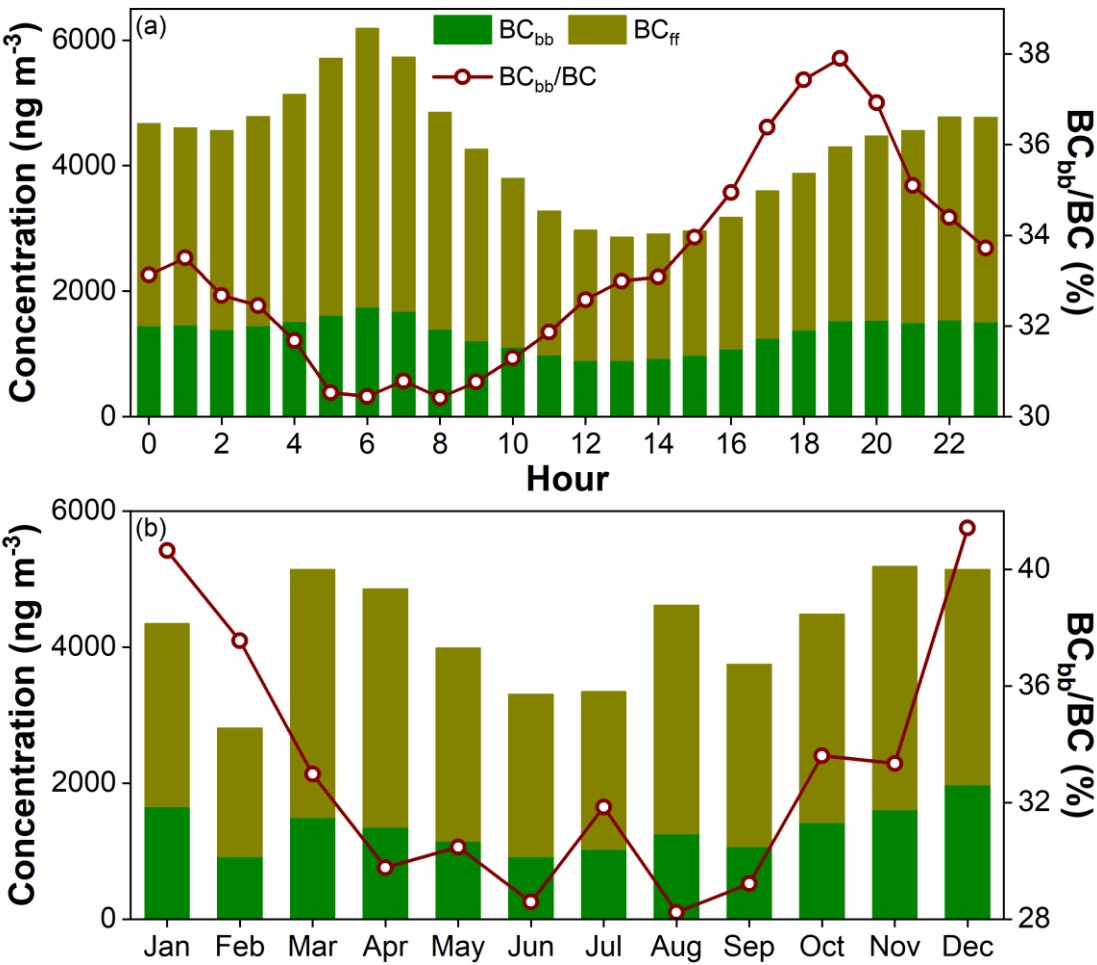

**Figure 5. Diurnal and monthly variations in BC<sub>ff</sub> and BC<sub>bb</sub> concentrations with the BC<sub>bb</sub>/BC fraction.**

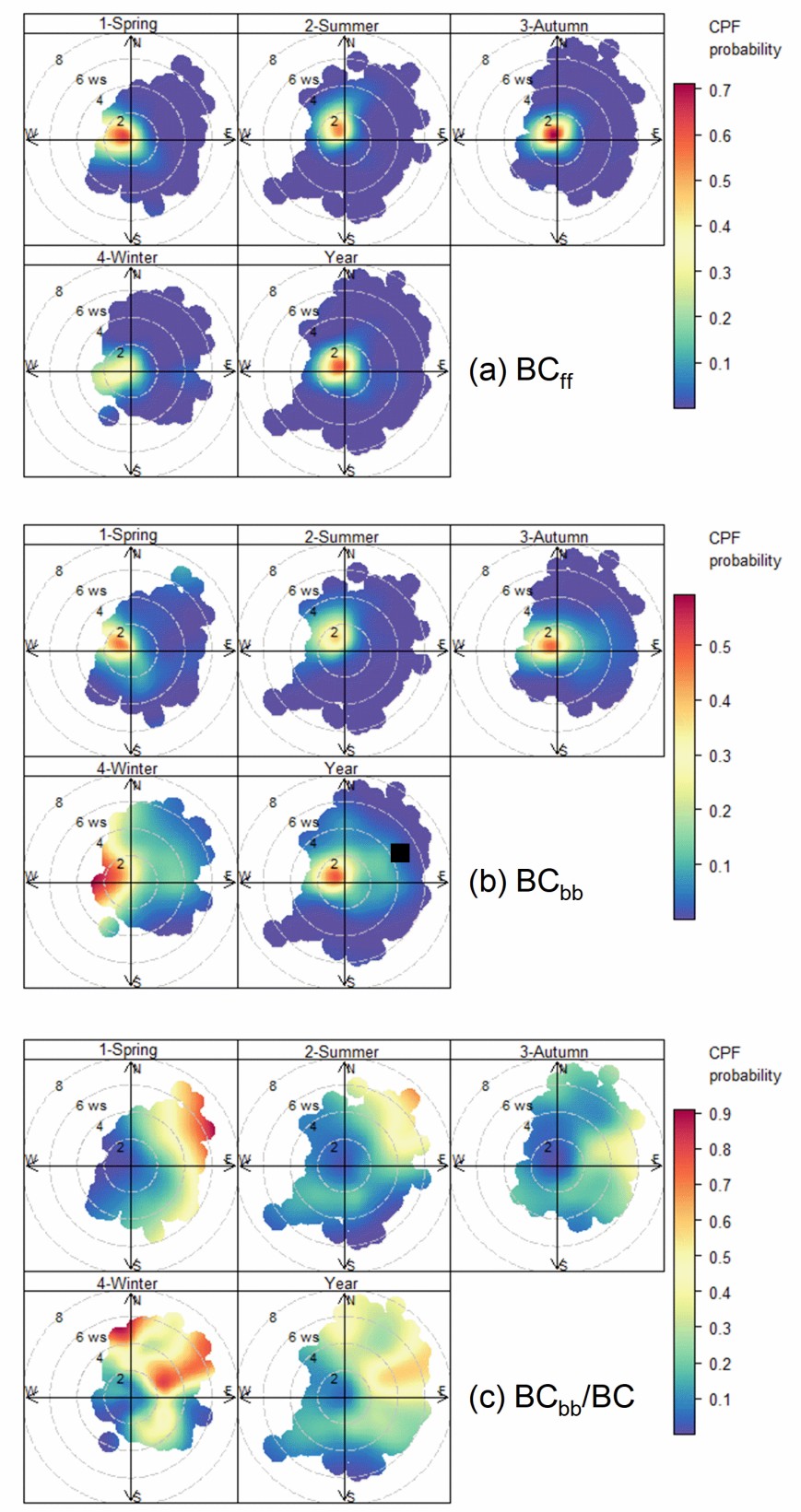

**Figure 6. CPF plots for (a) BC$_{ff}$, (b) BC$_{bb}$ and (c) BC$_{bb}$ contribution in Xiamen in 2014. ws represents wind speed (m s$^{-1}$).**

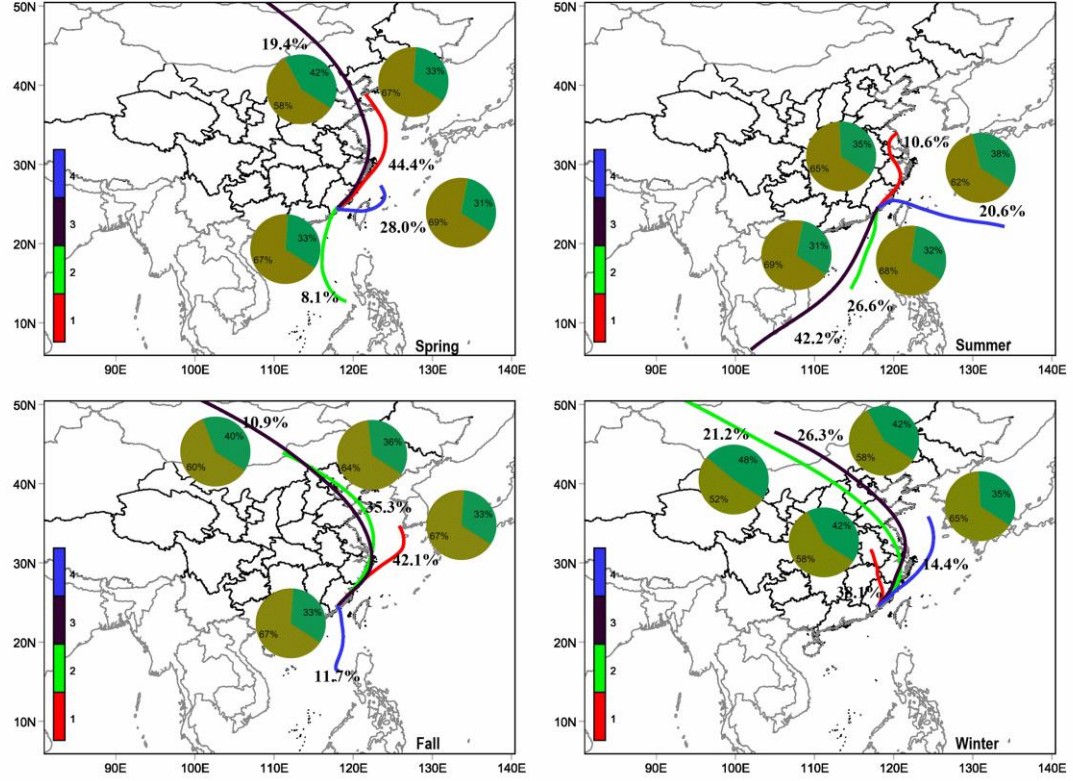

**Figure 7. Seasonal cluster-mean of five-day backward trajectories at 500 m with the corresponding trajectory percentages and BC$_{bb}$ contributions in Xiamen. The pie charts represent relative contributions of BC$_{ff}$ (dark yellow) and BC$_{bb}$ (olive green). The four-colored legend indicates the four different trajectory clusters. The percentage along the trajectories represent the percentage of each cluster in all trajectories.**

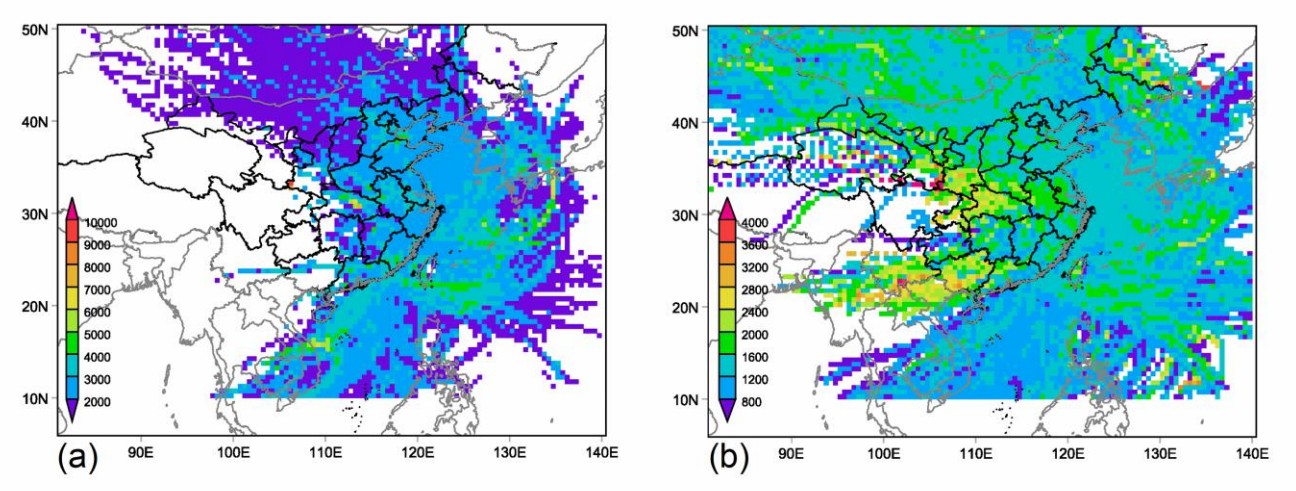

**Figure 8. Concentration-weighted trajectory (CWT) maps (ng m$^{-3}$) for (a) BC$_{ff}$ and (b) BC$_{bb}$ in Xiamen in 2014.**

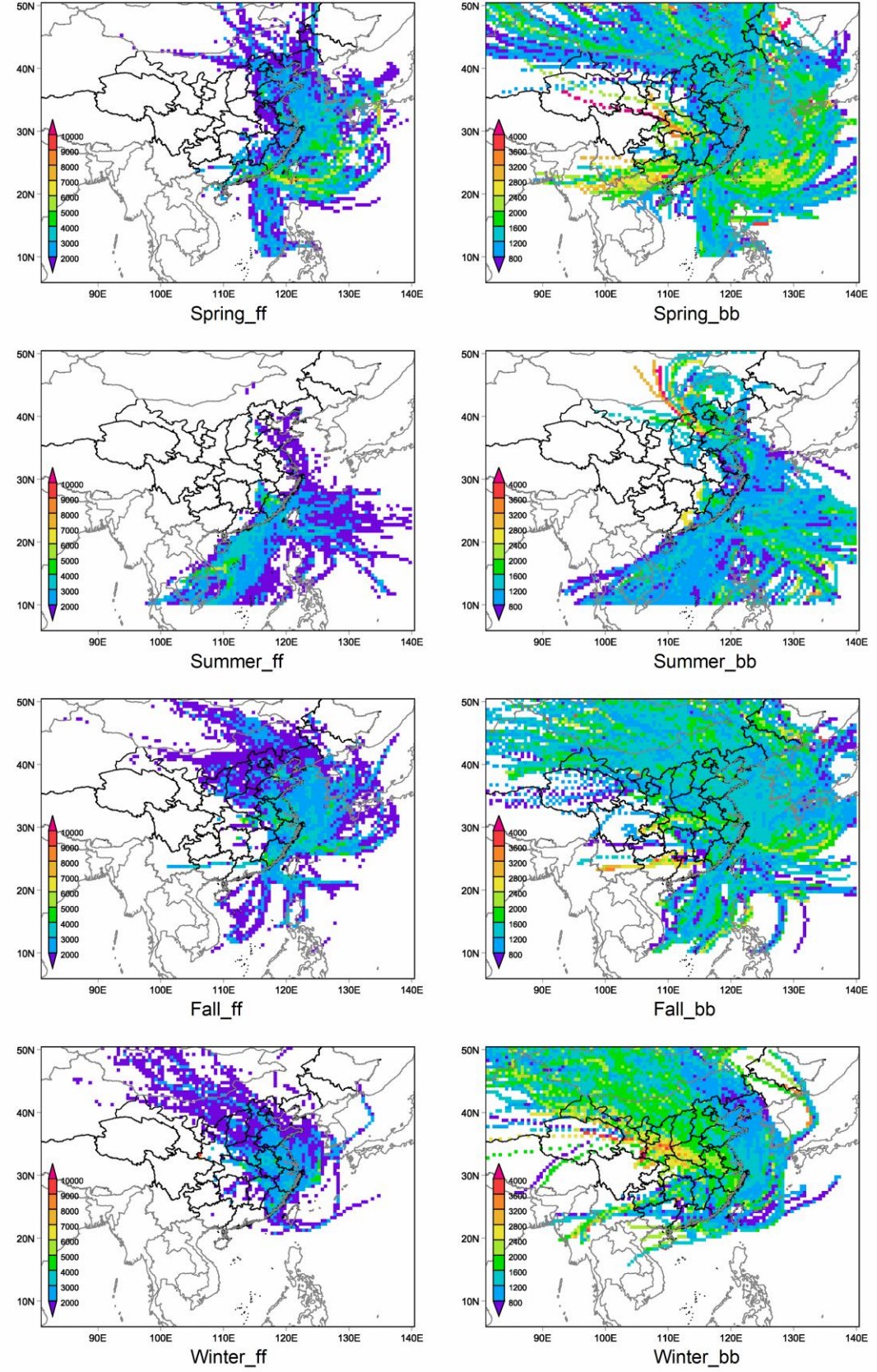

735

**Figure 9. Concentration weighted trajectory (CWT) maps for BC$_{ff}$ and BC$_{bb}$ in Xiamen for different seasons in 2014.**

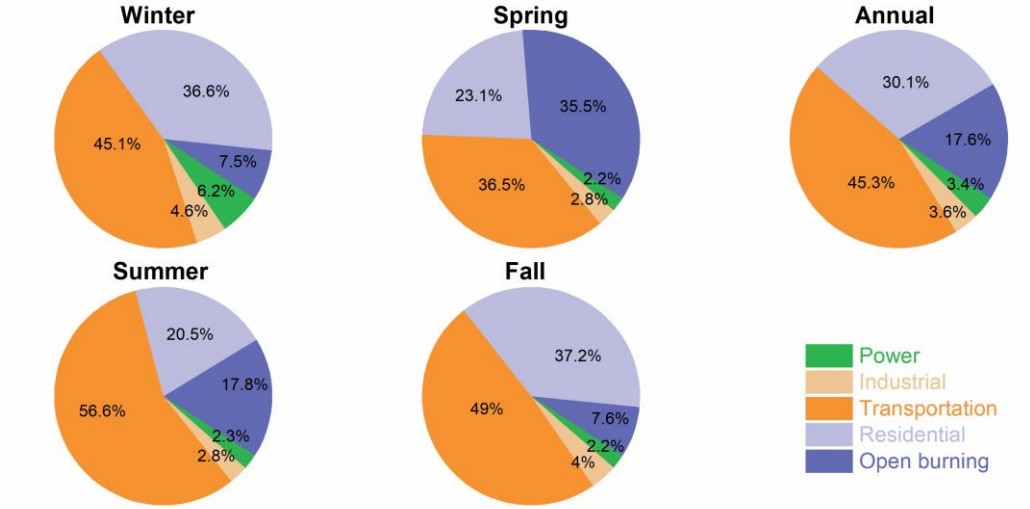

**Figure 10. Source contributions to BC of five source sectors in each period based on source-oriented model.**

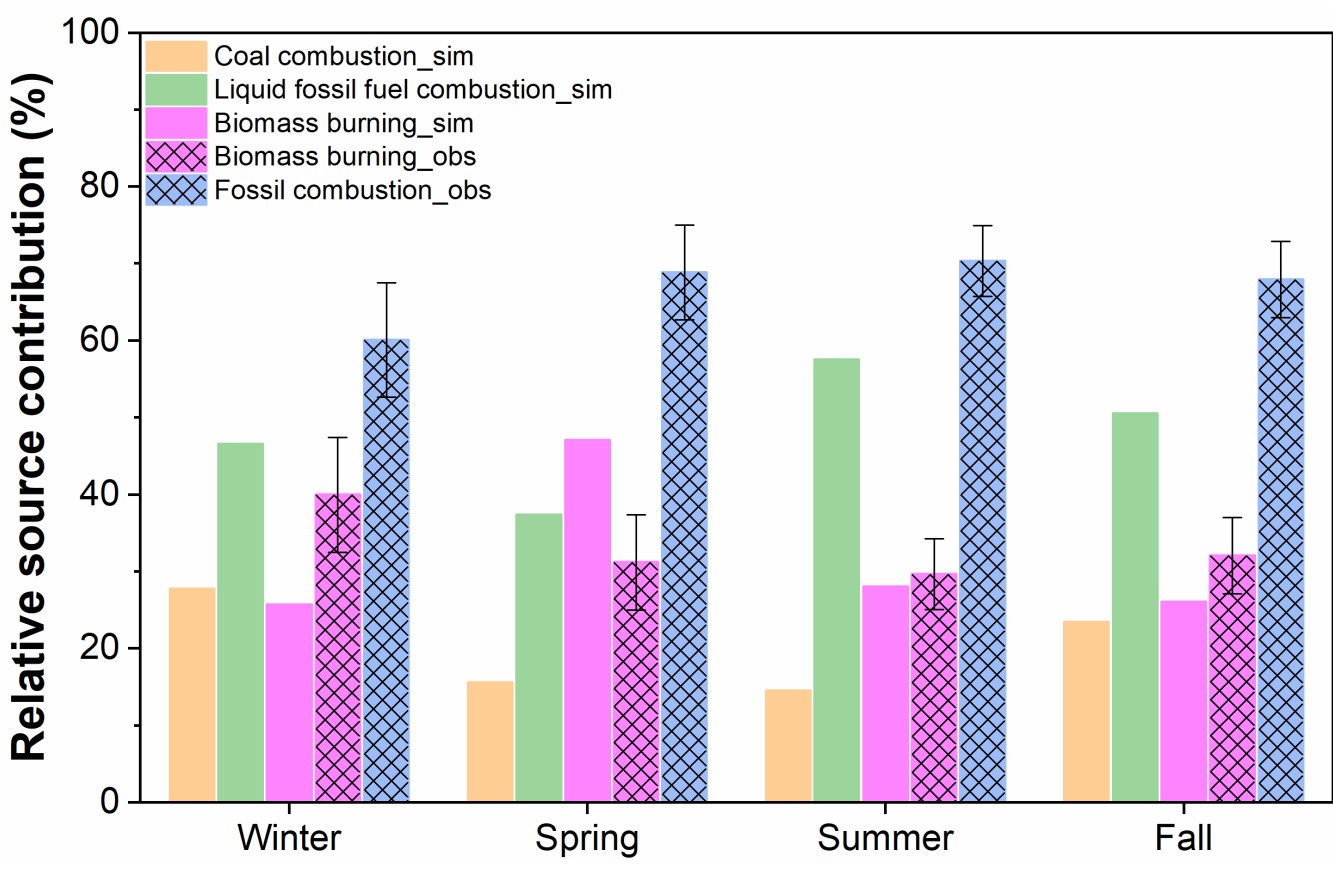

**Figure 11. Comparison of seasonal source contribution to BC between source-oriented model and Aethalometer method.**