# Peer review of "Source apportionment of black carbon aerosols from light absorption observation and source-oriented modeling: An implication in a coastal city in China"

_Atmospheric Chemistry and Physics, 2020_

## Referee Comment (RC1) · Anonymous Referee #1 · 17 Jun 2020

**Independent scientific review of "Source apportionment of black carbon aerosols from light absorption observation and source-oriented modeling: An implication in a coastal city in China" by Deng et al. 2020.**

**General**

This paper presents a comparison and analysis of BC in the Xiamen region in China, a cleaner region than others from what I understand. The region is mostly affected by long range transport of pollutants. The authors have utilized light absorption measurements from the aethalometer (AE31-model) and further applied the source apportionment method denoted as the aethalometer model. Two BC fractions originating from biomass burning (bb) and fossil fuel (ff) combustion where derived from the model and further compared with data from trajectory analysis, satellite observations and emission inventory modelling.

The novelty of the paper lies in the methodology, i.e. the comparison of the emission inventory model data to observations. Although, I don't agree with the authors that the model were in agreement with the observations, the chosen methodology is conducted in a relatively clean region in China and pin-points the importance of comprehensive and correct emission inventories.

The written language is on an average level. Some small errors occur occasionally which I haven't bothered to comment on.

I would recommend this manuscript for publication in ACP after the following comments have been considered.

**Major comments**

My largest concern with this paper is that the BCff analysis and comparison to the CMAQ-model is lacking. A long piece of text is written to compare the BCbb to the CMAQ-model results, why is the BCff-comparison omitted?

The chosen AAE for the aethalometer model is very arbitrary selected. A lot has happened since the Sandradewi paper from 2008. Zotter et al. (2017) gave a comprehensive analysis of AAE selection. The authors may also consider conducting their own selection, perhaps based on a literature review.

The authors claim that the CMAQ-model performed well and where in good agreement with the BC observations. How is this even possible to say? Studying Fig. 10, there is a considerable difference between the observations of BCbb and modeled BCbb, especially during winter and spring. A statistical test, i.e. t-test or similar, would enable the authors to make a justified statement regarding the agreement. And, again the comparison to observed BCff is lacking in Fig. 10 and totally omitted in the text.

**Minor comments**

L. 103. Please state the inlet of the AE31 as well as meters above sea level for the station.

L. 115. Which MAC-values were used for the BC calculation?

L. 130-132. Selection of AAE, please see my comment above.

L. 174. CMAQ is an abbreviation I suppose, for what?

L. 216-218. The PM2.5 was measured with which instrument?

L. 216-219. This is a very strange sentence, which I don't understand with all these parentheses and all. Please rewrite.

L. 215-225. Please explain clearly what you mean with high- and low-BC episodes.

L. 220-225. Why not include Fig. S2 in the main manuscript?

L. 239. How can BCbb be influenced by traffic?

L. 241.  How can the ratio be maximized when there is an evening rush-hour? I guess it is maximized due to increases in BB activity.

L. 253. I do understand that there is increased activity of domestic burning during winter. But why an increase in open-field biomass burning?

L. 257. Monsoon means more rain, how can there be increased burnings?

L. 310. BCbb from the sea? Please explain this.

L. 318. Please explain the open burning activity.

L. 329. Why does this activity exist in this region during this period of time?

L. 345-350. I would also like to see a similar analysis and comparison for the BCff and its potential sources. Why omitting it?

Figure 5. Please add more information in the Figure caption. The pie charts represents what? Why are there four trajectories. The four-coloured legend represents what? The percentage along the trajectories represents what?

Figure 6. Fig. 6. c and d should have the same scale on the legend for visual comparison.

Figure 10. Please include the fossil fuel combustion observations in this graph (i.e. BCff). There are no figure 10 b as stated in the figure caption.

---

## Referee Comment (RC2) · Anonymous Referee #2 · 30 Jun 2020

The manuscript by Deng at al. compared the source contributions from observation and CMAQ based source apportionment. Results indicate that while the BC predictions by the chemical transport model is an accordance with observations, a difference in source contributions can be observed. Overall, the manuscript is well written and addresses pertinent issues related to BC sources in a coastal city of China. To my knowledge this is one of the few studies which compare BC source apportionment using a source oriented model and observation based data. Such studies are needed as they directly evaluate the emission inventory of a region, with which many analysis and

action plans to reduce air pollution are formulated. Comments: 1) As the observation data was collected in a single location, how reasonable it is to compare the results with a model simulation of 36X36 Km. In other words, why was a finer domain not selected? At least the authors should discuss the influence of grid resolution on the performance. 2) How reasonable is it to take 500 m as the back trajectory height in all seasons? Shouldn't it be a function of mixing layer height? 3) More details about the cluster analysis carried out can be included. 4) From the CPF analysis, the potential sources to BC in different seasons can be identified, are they similar to the ones in emission inventory? 5) As the idea from back trajectory analysis is to qualitatively discuss local vs regional influence. It would be better if the main manuscript has only CWT and PSCF can be moved to supplementary. 6) What's the source of BC from a sea? Line 310 reads "The South China Sea were the main potential source for BCff and BCbb during summer." 7) In Figure 9, the methodology used for source apportionment should be included in the title.

---

## Referee Comment (RC3) · Anonymous Referee #4 · 2 Jul 2020

As BC aerosols play an important role in climate change and haze pollution, it has been a hot topic to identify, source apportion, and locate their sources. The main methods include observation based receptor modeling, emission inventory, atmospheric transport simulation. This study integrated the observation based modeling and emission inventory based transportation modeling to provide a more comprehensive picture regarding BC sources in a southeast coastal city. It clearly showed advancement comparing to similar studies in this topic and merit publication. Below are some comments that I would like the authors to address to improve its scientific quality.

1. Lines 125 and 131: the authors assumed AAE values for fossil fuel BC and biomass BC (i.e., 1 and 2), which play an important role in Aethalometer model for apportioning BC_ff and BC_bb. Although the authors provided references supporting their chosen values for this parameter, there are more recent studies regarding BC's AAE which show a quite wide range of 0.6-1.3 (Liu et al, ACP 18, 6259-6273, 2018). I would suggest to do an uncertainty and sensitivity analysis to investigate the impact of AAE on their results and provide a range of apportionment results instead of a certain value based on AAE = 1 or 2. Reference: Chao Liu, Chu Eddy Chung, Yan Yin, and Martin Schnaiter, The absorption Ångström exponent of black carbon: from numerical aspects, Atmos. Chem. Phys., 18, 6259–6273, 2018.

2. Line 134: "470nm was selected as near-ultraviolet wavelength" According to light spectrum, 470nm is blue light (450nm-490nm) within the visible light spectrum. Near UV is 300-400nm.

3. Line 140: nïĄĎïĄś is the total occurrences from wind sector ïĄĎïĄś. Did you consider the duration of the occurrence? For example, there were two occurrences that the wind blew from 45 degree. One lasted for 1 sec and the other one lasted for 1 min, during which BC exceeded the threshold criterion. If you use occurrence in equation 5, the CPF is 50%. If you use duration in equation 5, the CFP is 98.4%. Which method do you think makes more sense?

4. Line 143: top 25% concentration was chosen as the threshold criterion. I did not see justification or reference for doing so.

5. Lines 240-241: "BC_bb fraction is lowest at 8am and INCREASE due to the decrease in traffic emission. If this is true, BC_bb should DECREASE when the traffic emission increase. Why at 19:00 rush hour BC_bb fraction reaches the highest instead of lowest, when the traffic emission peaks?

6. Lines 341-342: you see the discrepancy between the observation based results and modeling results and try to explain why modeling underestimate BC_bb in winter. It seems you have subjective preference for observation method over modeling method. As we know both methods have uncertainties and nobody really know what the true FF/BB apportionment is. In winter, Aetholometer method yielded larger BC_bb fraction than modeling method but it doesn't necessarily mean the modeling "underestimate" the BC_bb fraction. Could it be that the Aetholometer method "overestimate", or both overestimate but Aetholometer overestimate more, or both underestimate but Aethlometer underestimate less? I'd like to see more in-depth investigation regarding the discrepancy and an objective, comprehensive discussion of both observation and modeling results, instead of just focusing on the issues of modeling method.

---

## Referee Comment (RC4) · Anonymous Referee #3 · 3 Jul 2020

Review of the manuscript "Source apportionment of black carbon aerosols from light absorption observation and source-oriented modeling: An implication in a coastal city in China" by Deng et al. Black carbon (BC) aerosol has significant influence on regional air quality and climate changes. However, uncertainties of the BC radiative forcing and climate effects still exist due to lack of observational understanding on BC sources, and subsequently optical properties. This manuscript compared the source contributions of BC in a relatively clean region in China from light-absorption based observation and source-oriented CMAQ model, and analyzed their temporal variations and spatial

originations. In my opinion, the manuscript presents a good work since the studies comparing different source apportionment techniques of BC are limited, although it is of great importance. The novelty is very good and the manuscript is well organized. I believe this manuscript could be accepted for publication in ACP after my following concerns are addressed. Major comments 1. This work compares the light-absorption based method with the source-oriented CMAQ model and the agreement between the two methods seems acceptable according to the comparison. However, the model performance on BC was not mentioned. I believe it is very important to add discussion about the evaluation of the model with the BC observations to make the comparison more reliable. 2. It is important to clarify the differences of the source-oriented modeling and other methods in determining BC sources such as brute force and PSAT. 3. When separating BCbb and BCff with the light-absorption of BC, why do the authors select 470 and 950 nm? Some studies adopted other wavelength combination. I suggest the authors comparing their selection with other combination to evaluate the impacts on the source apportionment results. 4. How are babs, ff and babs, bb calculated? It's not clearly enough in the manuscript. Minor comments 1. L174: Define the abbreviation CMAQ when it appeared for the first time. 2. L310: Why the South China Sea was source of BC? 3. Some spelling/grammar mistakes should be corrected.

---

## Author Comment (AC1) · 17 Oct 2020

Author Responses to Referees' Comments on **"Source apportionment of black carbon aerosols from light absorption observation and source-oriented modeling: An implication in a coastal city in China"** by J. Deng et al. (MS No.: acp-2020-451)

We appreciate the valuable comments and suggestions from the reviewers. Our point-by-point responses are provided below. The referees' comments are in black and our responses are highlighted in red.

**Responses to Referee #1**

**General**

This paper presents a comparison and analysis of BC in the Xiamen region in China, a cleaner region than others from what I understand. The region is mostly affected by long range transport of pollutants. The authors have utilized light absorption measurements from the aethalometer (AE31-model) and further applied the source apportionment method denoted as the aethalometer model. Two BC fractions originating from biomass burning (bb) and fossil fuel (ff) combustion where derived from the model and further compared with data from trajectory analysis, satellite observations and emission inventory modelling.

The novelty of the paper lies in the methodology, i.e. the comparison of the emission inventory model data to observations. Although, I don't agree with the authors that the model were in agreement with the observations, the chosen methodology is conducted in a relatively clean region in China and pin-points the importance of comprehensive and correct emission inventories. The written language is on an average level. Some small errors occur occasionally which I haven't bothered to comment on. I would recommend this manuscript for publication in ACP after the following comments have been considered.

Response:

We would like to express our sincere thanks to the reviewer for the constructive and valuable comments, which helped us to effectively improve the quality of this paper. Besides, we accordingly revised some mistakes and improved the written English throughout the manuscript. Please see our itemized responses below.

**Major comments**

My largest concern with this paper is that the $BC_{ff}$ analysis and comparison to the CMAQ-model is lacking. A long piece of text is written to compare the $BC_{bb}$ to the CMAQ-model results, why is the $BC_{ff}$-comparison omitted?

Response:

In both the Aethalometer method and source-oriented CMAQ model, total BC was divided into two components, which were BC from fossil-fuel combustion (i.e. $BC_{ff}$) and biomass burning (i.e. $BC_{bb}$). If the observed $BC_{bb}$ contribution was smaller than the simulated $BC_{bb}$ contribution, as a matter of course, the observed $BC_{ff}$ contribution would be larger than the simulated $BC_{ff}$ contribution. And vice versa. Therefore, in the original manuscript, we only compared $BC_{bb}$ contribution from the two methods and omitted the $BC_{ff}$-comparison.

According to this comment, the observed relative contribution of $BC_{ff}$ was implemented in Fig. 10, and $BC_{ff}$ analysis and comparison was also added in the revision.

The chosen AAE for the aethalometer model is very arbitrary selected. A lot has happened since the Sandradewi paper from 2008. Zotter et al. (2017) gave a comprehensive analysis of AAE selection. The

authors may also consider conducting their own selection, perhaps based on a literature review.

Response:

Thanks for the valuable and helpful suggestion. In previous studies, fixed AAE values of fossil fuel ($\alpha_{ff}$) and biomass burning ($\alpha_{bb}$) were usually used in source apportionment studies for simplicity (Favez et al., 2010; Herich et al., 2011; Sciare et al., 2011; Harrison et al., 2013; Fuller et al., 2014; Rajesh and Ramachandran, 2017; Zotter et al., 2017; Helin et al., 2018). Therefore, one of the largest uncertainties in the Aethalometer model comes from the preselected $\alpha_{ff}$ and $\alpha_{bb}$ values (Sciare et al., 2011; Healy et al., 2017; Zotter et al., 2017; Helin et al., 2018). Zotter et al. (2017) determined the site specific $\alpha_{ff}$ and $\alpha_{bb}$ values by using auxiliary measurements (e.g. EC/OC, and $^{14}$C measurements). Some other work estimated site specific $\alpha_{ff}$ and $\alpha_{bb}$ values using levoglucosan measurements together with the Aethalometer model (Fuller et al., 2014; Titos et al., 2017). In addition, many published studies commonly used values in the literature (Elser et al., 2016; Healy et al., 2017; Martinsson et al., 2017; Rajesh and Ramachandran, 2017; Xiao et al., 2020).

Based on a comprehensive literature review (Table S1 in the revision), we found that $\alpha_{ff}$ and $\alpha_{bb}$ values were most commonly in the range of 0.9–1.1 and 1.7–2.2, respectively. Due to the lack of supporting measurements (e.g. $^{14}$C measurements, and chemical samples) in this work, the empirical value of $\alpha_{ff}$ and $\alpha_{bb}$ is adopted as 1.0 and 2.0 following many previous studies (Kirchstetter et al., 2004; Favez et al., 2010; Crippa et al., 2013; Fuller et al., 2014; Crilley et al., 2015; Petit et al., 2017; Xiao et al., 2020). The performance of the "Aethalometer model" is tested by sensitivity analysis using various combinations of AAE pairs for the assessment of $BC_{ff}$ and $BC_{bb}$. Therefore, except of the initial estimates with $\alpha_{ff}$=1.0 and $\alpha_{bb}$=2.0, the analysis was repeated by changing the $\alpha_{ff}$ from 0.9 to 1.1 and the $\alpha_{bb}$ from 1.7 to 2.2.

The above statements (Line 140-155) and the sensitivity analysis (Line 250-259) were added in the revision.

The authors claim that the CMAQ-model performed well and where in good agreement with the BC observations. How is this even possible to say? Studying Fig. 10, there is a considerable difference between the observations of $BC_{bb}$ and modeled $BC_{bb}$, especially during winter and spring. A statistical test, i.e. t-test or similar, would enable the authors to make a justified statement regarding the agreement. And, again the comparison to observed $BC_{ff}$ is lacking in Fig. 10 and totally omitted in the text.

Response:

We agree with the reviewer that there are considerable differences between the BC source apportionment results from the Aethalometer method and source-oriented CMAQ model in winter (14.3%) and spring (15.9%). We believe that the uncertainties in the Aethalometer method and the emission inventory used in the source-oriented model may lead to the unignorable gap between observation-based and model results. However, Fig. 10 also shows the discrepancies between the results from the two source apportionment methods are much smaller in summer (1.7%) and fall (6.0%). For the entire year, the annual average relative contributions of $BC_{ff}$ and $BC_{bb}$ derived by the source-oriented model are 67.4% and 32.6%, which are very close to those (i.e., 66.7% for $BC_{ff}$ and 33.3% for $BC_{bb}$) obtained by the Aethalometer method. In the revision, we accordingly revised our statements. In addition, we implemented the observed contribution of $BC_{ff}$ in Fig. 10, and added $BC_{ff}$ analysis and comparison in the revision based on the reviewer's suggestion.

**Minor comments**

L. 103. Please state the inlet of the AE31 as well as meters above sea level for the station.

Response:

The AE31 is with a $PM_{2.5}$ cut-off inlet. The station is ~10 m a.s.l. The statements have been revised (Line 105-110).

L. 115. Which MAC-values were used for the BC calculation?

Response:

The MAC value at 880 nm wavelength was 10.1 $m^2\ g^{-1}$ based on our previous study (Deng et al., 2016).

Reference:

Deng, J., Zhang, Y., Hong, Y., Xu, L., Chen, Y., Du, W., and Chen, J.: Optical properties of PM2.5 and the impacts of chemical compositions in the coastal city Xiamen in China, *Sci. Total Environ.*, 557–558, 665–675, 2016.

L. 130-132. Selection of AAE, please see my comment above.

Response:

Thanks for this comment. Please see our response above.

L. 174. CMAQ is an abbreviation I suppose, for what?

Response:

CMAQ is the abbreviation for Community Multiscale Air Quality Modeling System. We have added it in the text (Line 196-197).

L. 216-218. The $PM_{2.5}$ was measured with which instrument?

Response:

Mass concentration of $PM_{2.5}$ was synchronously measured by a Tapered Element Oscillating Microbalance (TEOM) sampler (RP1400, Thermo Fisher Scientific). We have added the statement in the revision (Line 262).

L. 216-219. This is a very strange sentence, which I don't understand with all these parentheses and all. Please rewrite.

Response:

We have rewritten the sentence as follows (Line 263-265):

"High-pollutant periods are the days with daily average concentration higher than the seasonal average plus one standard deviation, while low-pollutant periods are the days with daily average concentration lower than the seasonal average minus one standard deviation."

L. 215-225. Please explain clearly what you mean with high- and low-BC episodes.

Response:

Here, high-BC periods are the days with daily average BC concentration higher than the seasonal average plus one standard deviation (average + σ), and low-BC periods are the days with daily average BC concentration lower than the seasonal average minus one standard deviation (average - σ).

L. 220-225. Why not include Fig. S2 in the main manuscript?

Response:

Thanks for the suggestion. Accordingly, we moved Fig. S2 to the main manuscript as Fig. 4.

L. 239. How can $BC_{bb}$ be influenced by traffic?

Response:

The sentence has been revised to "$BC_{bb}$ was not influenced by traffic-related emission".

L. 241. How can the ratio be maximized when there is an evening rush-hour? I guess it is maximized due to increases in BB activity.

Response:

We agree with you that the maximum $BC_{bb}$ fraction toward evening is due to increase in biomass burning activity, such as domestic biofuel burning for cooking. We have revised the statement.

L. 253. I do understand that there is increased activity of domestic burning during winter. But why an increase in open-field biomass burning?

Response:

He et al. (2011) determined the seasonal variations in the open-filed biomass burning, including crop-residual burning and forest fire, by using the fire-counts data. They found that most of the fire hotspots around South China were detected in winter months since the period is typically on the harvest and dry season.

Reference:

He, M., Zheng, J., Yin, S., and Zhang, Y.: Trends, temporal and spatial characteristics, and uncertainties in biomass burning emissions in the Pearl River Delta, China, *Atmos. Environ.*, 45 (24), 4051–4059, 2011.

L. 257. Monsoon means more rain, how can there be increased burnings?

Response:

Here we do not mean that summer monsoon leads to biomass burning. In our previous study, we found that intensive open biomass burning occurred in Southeast Asia in July (Qiu et al., 2019). Under the control of summer monsoon, the southerly wind would take air pollutants emitted from biomass burning in Southeast Asia to Southeast China.

Reference:

Qiu, Y., Wu, X., Zhang, Y., Xu, L., Hong Y., Chen, J., Chen, X., and Deng, J.: Aerosol light absorption in a coastal city in Southeast China: temporal variations and implications for brown carbon, *J. Environ. Sci.*, 80, 257–266, 2019.

L. 310. $BC_{bb}$ from the sea? Please explain this.

Response:

Thanks for pointing out this mistake. In this study, the East China Sea and South China Sea were indicated as the potential source for BC. However, these areas should not be actual source areas and might be identified due to the trailing effect (Lee et al., 2014; Deng et al., 2020). We have corrected the statements in the revision (Line 361-363).

References:

Deng, J., Zhao, W., Wang, X., Wu, L., Hu, W., Ren, L., and Fu, P.: Black carbon in Xiamen, China: temporal variations, transport pathways and impacts of synoptic circulation, *Chemosphere*, 241, 125133, 2020.

Lee, G., Kim, P., Han, Y., Holsen, T., and Lee, S.: Tracing sources of total gaseous mercury to Yongheung Island off the coast of Korea, *Atmosphere*, 5, 273–291, 2014.

L. 318. Please explain the open burning activity.

Response:

Open burning activity (i.e., open biomass burning) generally refers to open combustion of various biomass materials such as forest vegetation, crop residue and municipal solid waste (Permadi and Oanh, 2013). We have added the statements in the revision (Line 234).

Reference:

Permadi, D. A., and Oanh, N. T. K.: Assessment of biomass open burning emissions in Indonesia and potential climate forcing

impact, *Atmos. Environ.*, 78, 250–258, 2013.

L. 329. Why does this activity exist in this region during this period of time?

Response:

South Asia (Bangladesh, Bhutan, India, Nepal, and Pakistan) and Southeast Asia are two regional hot spots for open biomass burning (Singh et al., 2020). Myanmar in Southeast Asia is the biggest source of forest fires among Asian countries during the spring months of March and April (Vadrevu et al., 2015). Emissions from agricultural residue burning during April–May over the Indo-Gangetic Plain (IGP) causes a serious air quality problem (Sharma et al., 2010). Some relevant references have been added in the revision.

References:

Sharma, A. R., Kharol, S. K., Badarinath, K. V. S., and Singh, D.: Impact of agriculture crop residue burning on atmospheric aerosol loading - A study over Punjab State, India, *Ann. Geophys.*, 28, 367–379, 2010.

Singh, P., Sarawade, P., and Adhikary, B.: Carbonaceous Aerosol from Open Burning and its Impact on Regional Weather in South Asia, *Aerosol Air Qual. Res.*, 20, 419–431, 2020.

Vadrevu, K. P., Lasko, K., Giglio, L., and Justice, C.: Vegetation fires, absorbing aerosols and smoke plume characteristics in diverse biomass burning regions of Asia, *Environ. Res. Lett.*, 10, 105003, 2015.

L. 345-350. I would also like to see a similar analysis and comparison for the BCff and its potential sources. Why omitting it?

Response:

According to the suggestion, we added a similar analysis and brief comparison for BCff.

Figure 5. Please add more information in the Figure caption. The pie charts represents what? Why are there four trajectories. The four-coloured legend represents what? The percentage along the trajectories represents what?

Response:

We revised the figure caption by adding more information according to the suggestion. In the figure, the pie charts represent relative contributions of BCff (dark yellow) and BCbb (olive green). Four trajectory-clusters were obtained for each season with the clustering option of Euclidean distance. Cluster analysis was performed with the TrajStat software and "Eye ball" method was used to determine the cluster number in this software. The four-coloured legend indicates the four different trajectory clusters. The percentage along the trajectories represent the percentage of each cluster in all trajectories.

Figure 10. Please include the fossil fuel combustion observations in this graph (i.e. BCff). There are no Figure 10b as stated in the figure caption.

Response:

Thanks for the reminder. According the suggestion, we added the fossil fuel combustion observations into Fig. 10 and corrected the figure caption.

**Responses to Referee #2**

The manuscript by Deng at al. compared the source contributions from observation and CMAQ based source apportionment. Results indicate that while the BC predictions by the chemical transport model is an accordance with observations, a difference in source contributions can be observed. Overall, the manuscript

is well written and addresses pertinent issues related to BC sources in a coastal city of China. To my knowledge this is one of the few studies which compare BC source apportionment using a source oriented model and observation based data. Such studies are needed as they directly evaluate the emission inventory of a region, with which many analysis and action plans to reduce air pollution are formulated.

Response:

We appreciate the reviewer's positive remarks on the importance of the work. We have revised the manuscript according to the reviewers' comments to improve its quality.

**Comments:**

1) As the observation data was collected in a single location, how reasonable it is to compare the results with a model simulation of 36X36 Km. In other words, why was a finer domain not selected? At least the authors should discuss the influence of grid resolution on the performance.

Response:

Although the observation showed in this study is from one single location, the model has been validated against all available observations within China including gas species as well as $PM_{2.5}$ and its components (Hu et al., 2015, 2016). A finer resolution is a good idea but is limited by lacking of input information such as emission inventory.

As suggested by the reviewer, we added below discussion regarding the grid resolution (Line 410-412):

"In addition, low resolution of simulation could lead to uncertainties in model results. The peak values close to emission sources may not be captured after the dilution of emission in large grid cells. In future, high resolution is suggested as long as high-resolution emission inventories are available".

References:

Hu, J., Chen, J., Ying, Q., and Zhang, H.: One-year simulation of ozone and particulate matter in China using WRF/CMAQ modeling system, Atmos. Chem. Phys., 16, 10333–10350, 2016.

Hu, J., Wu, L., Zheng, B., Zhang, Q., He, K., Chang, Q., Li, X., Yang, F., Ying, Q., and Zhang, H.: Source contributions and regional transport of primary particulate matter in China, Environ. Pollut., 207, 31–42, 2015.

2) How reasonable is it to take 500 m as the back trajectory height in all seasons? Shouldn't it be a function of mixing layer height?

Response:

Using the height as a function of mixing layer height as the backward trajectory height may be a good idea. However, the mixing layer height varies over time, making it difficult to compare the backward trajectories in different periods. In this study, 500 m was adopted as the back trajectory height because it was usually within the mixing layer. Besides, the same back trajectory heights in all seasons help to compare the sources and transport pathways of BC during different periods. In addition, the selection of back trajectory height was in accordance with many previous studies (Sunnu et al., 2013; Su et al., 2015; Qiu et al., 2019; Deng et al., 2020).

References:

Deng, J., Zhao, W., Wang, X., Wu, L., Hu, W., Ren, L., and Fu, P.: Black carbon in Xiamen, China: temporal variations, transport pathways and impacts of synoptic circulation, *Chemosphere*, 241, 125133, 2020.

Qiu, Y., Wu, X., Zhang, Y., Xu, L., Hong Y., Chen, J., Chen, X., and Deng, J.: Aerosol light absorption in a coastal city in Southeast China: temporal variations and implications for brown carbon, *J. Environ. Sci.*, 80, 257–266, 2019.

Su, L., Yuan, Z., Fung, J. C. H., and Lau, A. K. H.: A comparison of HYSPLIT backward trajectories generated from two GDAS datasets, *Sci. Total Environ.*, 506–507, 527–537, 2015.

Sunnu, A., Resch, F., and Afeti, G.: Back-trajectory model of the Saharan dust flux and particle mass distribution in West Africa, *Atmos. Res.*, 9, 125–132, 2013.

3) More details about the cluster analysis carried out can be included.

Response:

Cluster analysis for the trajectories was performed based on the inputs of hourly backward trajectories with the TrajStat plugin of the Meteoinfo software. Four clusters were obtained for each season with the clustering option of Euclidean distance.

Above details have been included in the revision (Line 168-171).

4) From the CPF analysis, the potential sources to BC in different seasons can be identified, are they similar to the ones in emission inventory?

Response:

The CPF analysis was used to identify the possible predominant directions of local sources of BC. However, the emission inventory (MEIC) used in the source-oriented model was on a regional scale. Therefore, it is difficult to compare the potential sources from CPF analysis with the ones in emission inventory.

5) As the idea from back trajectory analysis is to qualitatively discuss local vs regional influence. It would be better if the main manuscript has only CWT and PSCF can be moved to supplementary.

Response:

Thanks for the suggestion. Accordingly, we moved the PSCF results to supplementary materials (Figure S2 and Figure S3).

6) What's the source of BC from a sea? Line 310 reads "The South China Sea were the main potential source for $BC_{ff}$ and $BC_{bb}$ during summer."

Response:

In this study, the East China Sea and South China Sea were indicated as the potential source for BC. However, these areas should not be actual source areas and might be identified due to the trailing effect (Lee et al., 2014; Deng et al., 2020). We have corrected the statements in the revision (Line 361-363).

References:

Deng, J., Zhao, W., Wang, X., Wu, L., Hu, W., Ren, L., and Fu, P.: Black carbon in Xiamen, China: temporal variations, transport pathways and impacts of synoptic circulation, *Chemosphere*, 241, 125133, 2020.

Lee, G., Kim, P., Han, Y., Holsen, T., and Lee, S.: Tracing sources of total gaseous mercury to Yongheung Island off the coast of Korea, *Atmosphere*, 5, 273–291, 2014.

7) In Figure 9, the methodology used for source apportionment should be included in the title.

Response:

Thanks for the suggestion. We have added the methodology in the title of Figure 9.

**Responses to Referee #3**

Black carbon (BC) aerosol has significant influence on regional air quality and climate changes. However, uncertainties of the BC radiative forcing and climate effects still exist due to lack of observational understanding on BC sources, and subsequently optical properties. This manuscript compared the source contributions of BC in a relatively clean region in China from light-absorption based observation and source-oriented CMAQ model, and analyzed their temporal variations and spatial originations. In my

opinion, the manuscript presents a good work since the studies comparing different source apportionment techniques of BC are limited, although it is of great importance. The novelty is very good and the manuscript is well organized. I believe this manuscript could be accepted for publication in ACP after my following concerns are addressed.

Response:

Thanks a lot for the positive comments and helpful suggestions. We have revised the manuscript according to the reviewers' comments to improve its quality.

**Major comments**

1. This work compares the light-absorption based method with the source-oriented CMAQ model and the agreement between the two methods seems acceptable according to the comparison. However, the model performance on BC was not mentioned. I believe it is very important to add discussion about the evaluation of the model with the BC observations to make the comparison more reliable.

Response:

Thanks for the suggestion. Actually, in this study, the model performance on BC was conducted by comparing our measurements with the model results. As shown in section 3.3 and Fig. 10, this source-oriented CMAQ model almost captured the seasonal trends in BC from fossil fuel and biomass burning. Besides, the model has been validated against observations within China in our previous study (Hu et al., 2015). In that study, we compared the model results with the observations, finding that the source-oriented model could well reproduce the BC concentrations and there was a good agreement between the BC observation and simulation. Thus, we believe the model results of BC concentration are reliable.

Above discussion about the evaluation of the model has been added in the revision.

Reference:

Hu, J., Wu, L., Zheng, B., Zhang, Q., He, K., Chang, Q., Li, X., Yang, F., Ying, Q., and Zhang, H.: Source contributions and regional transport of primary particulate matter in China, *Environ. Pollut.*, 207, 31–42, 2015.

2. It is important to clarify the differences of the source-oriented modeling and other methods in determining BC sources such as brute force and PSAT.

Response:

PSAT (i.e. Particulate matter Source Apportionment Technology) is similar to the sourced-oriented modeling with tagged tracers. However, PSAT does not track the species from different sources directly in each time step. Instead, it allocates the changes of bulk concentrations to different sources after each time step based on the ratio of each source to total emissions.

Comparing with the source-oriented model, the brute force method (BFM) is more suitable to estimate the change of PM due to proposed emission control measures than to determine the contributions of certain sources because removal of PM emissions could affect the transport, chemistry, deposition and interactions with meteorology although they are not chemically reactive (Zhang and Ying, 2011). The results simulated with the BFM is different from "source apportionment" since the summation of the contributions of all source categories will not always equal the total concentration. In addition, the BFM needs to repeat chemical transport model simulations multiple times and greatly increases the computational cost.

The differences of the source-oriented modeling and other methods such as brute force and PSAT have been clarified in the revision.

Reference:

Zhang, H., and Ying, Q.: Contributions of local and regional sources of NOx to ozone concentrations in Southeast Texas, *Atmos. Environ.*, 45 (17), 2877–2887, 2011.

3. When separating BC$_{bb}$ and BC$_{ff}$ with the light-absorption of BC, why do the authors select 470 and 950 nm? Some studies adopted other wavelength combination. I suggest the authors comparing their selection with other combination to evaluate the impacts on the source apportionment results.

Response:

Thanks for the suggestion. In fact, several pairs of wavelengths (470/950 nm, 470/880 nm and 370/880 nm) have been previously used in literature. In this study, the combination of 470/950 nm was adopted according to the studies by Sandradewi et al. (2008), Favez et al. (2010), Zotter et al. (2017), Helin et al. (2018) and Kalogridis et al. (2018). According to the reviewer's suggestion, we compared our selection with the other combination of 370/880 nm. As shown in Figure R1, BC$_{ff}$ calculated with 470/950 nm was strongly correlated with BC$_{ff}$ calculated with 370/880 nm, with linear correlation coefficient $R^2$ of 0.996 and slope of 0.964. Similarly, BC$_{bb\_470/950}$ was also correlated with BC$_{ff\_370/880}$, with linear correlation coefficient $R^2$ of 0.963 and slope of 1.034. Such comparison results suggest that the impact of different combinations are acceptable and our results are reliable. Our finding is also supported by previous study of Garg et al. (2016), which indicated that there was no systematic bias introduced by any of the lamps.

Reference:

Garg, S., Chandra, B. P, Sinha, V., Sarda-Esteve, R., Gros, V., and Sinha, B.: Limitation of the use of the absorption angstrom exponent for source apportionment of equivalent black carbon: a case study from the north west indo-gangetic plain, *Environ. Sci. Technol.*, 50 (2),814–824, 2016.

[Figure]

Figure R1. Correlations between (a) BC$_{ff}$ and (b) BC$_{bb}$ calculated with the wavelength combinations of 470/950 nm and 370/880 nm.

4. How are $b_{abs,ff}$ and $b_{abs,bb}$ calculated? It's not clearly enough in the manuscript.

Response:

$b_{abs,ff}$ and $b_{abs,bb}$ are calculated with the method provided by Zotter et al. (2017).

The spectral dependence of the absorption is described by the power law $b_{abs}(\lambda) \sim \lambda^{-AAE}$. Consequently, for a wavelength pair the following relation can be derived:

$$\frac{b_{abs}(\lambda_1)}{b_{abs}(\lambda_2)} = \left(\frac{\lambda_1}{\lambda_2}\right)^{-\alpha} \tag{1}$$

In addition, $b_{abs}$ measured with the Aethalometer at different wavelengths can be apportioned into a biomass burning (bb) and a fossil fuel (ff) contribution. Therefore, $b_{abs,ff}$ and $b_{abs,bb}$ satisfy the following equation.

$$b_{abs}(\lambda) = b_{abs,ff}(\lambda) + b_{abs,bb}(\lambda) \tag{2}$$

Using Eqs. (1)-(2) and the measured babs at two different wavelengths, a fossil fuel and biomass burning

contribution can be apportioned using the following equations:

$$\frac{b_{abs,ff}\left(\lambda_{1}\right)}{b_{abs,ff}\left(\lambda_{2}\right)}=\left(\frac{\lambda_{1}}{\lambda_{2}}\right)^{-\alpha_{ff}} \tag{3}$$

$$\frac{b_{abs,bb}\left(\lambda_{1}\right)}{b_{abs,bb}\left(\lambda_{2}\right)}=\left(\frac{\lambda_{1}}{\lambda_{2}}\right)^{-\alpha_{bb}} \tag{4}$$

$$b_{abs,bb}=\frac{b_{abs}\left(\lambda_{1}\right)-b_{abs}\left(\lambda_{2}\right)\cdot\left(\frac{\lambda_{1}}{\lambda_{2}}\right)^{-\alpha_{ff}}}{\left(\frac{\lambda_{1}}{\lambda_{2}}\right)^{-\alpha_{bb}}-\left(\frac{\lambda_{1}}{\lambda_{2}}\right)^{-\alpha_{ff}}} \tag{5}$$

$$b_{abs,ff}=\frac{b_{abs}\left(\lambda_{1}\right)-b_{abs}\left(\lambda_{2}\right)\cdot\left(\frac{\lambda_{1}}{\lambda_{2}}\right)^{-\alpha_{bb}}}{\left(\frac{\lambda_{1}}{\lambda_{2}}\right)^{-\alpha_{ff}}-\left(\frac{\lambda_{1}}{\lambda_{2}}\right)^{-\alpha_{bb}}} \tag{6}$$

For given $\alpha_{ff}$ and $\alpha_{bb}$ values and $b_{abs}$ ($\lambda_1$) and $b_{abs}$ ($\lambda_2$), the values for $b_{abs,ff}$ and $b_{abs,bb}$ can be calculated by combining Eqs. (5) and (6).

The details have been added in the revision.

**Minor comments**

1. L174: Define the abbreviation CMAQ when it appeared for the first time.

Response:

We have revised it in the revision (Line 196) accordingly.

2. L310: Why the South China Sea was source of BC?

Response:

Thanks for pointing out this mistake. In this study, the East China Sea and South China Sea were indicated as the potential source for BC. However, these areas should not be actual source areas and might be identified due to the trailing effect (Lee et al., 2014; Deng et al., 2020). We have corrected the statements in the revision (Line 361-363).

References:

Deng, J., Zhao, W., Wang, X., Wu, L., Hu, W., Ren, L., and Fu, P.: Black carbon in Xiamen, China: temporal variations, transport pathways and impacts of synoptic circulation, *Chemosphere*, 241, 125133, 2020.

Lee, G., Kim, P., Han, Y., Holsen, T., and Lee, S.: Tracing sources of total gaseous mercury to Yongheung Island off the coast of Korea, *Atmosphere*, 5, 273–291, 2014.

3. Some spelling/grammar mistakes should be corrected.

Response:

According to the reviewers' suggestion, we checked the manuscript throughout and corrected some spelling and grammar mistakes to improve its quality.

**Responses to Referee #4**

As BC aerosols play an important role in climate change and haze pollution, it has been a hot topic to identify, source apportion, and locate their sources. The main methods include observation based receptor modeling, emission inventory, atmospheric transport simulation. This study integrated the observation based modeling and emission inventory based transportation modeling to provide a more comprehensive picture regarding BC sources in a southeast coastal city. It clearly showed advancement comparing to similar studies in this topic and merit publication. Below are some comments that I would like the authors to address to improve its scientific quality.

Response:

Your positive comments are greatly appreciated. We have revised the manuscript to improve its quality according to the comments from the reviewers.

Lines 125 and 131: the authors assumed AAE values for fossil fuel BC and biomass BC (i.e., 1 and 2), which play an important role in Aethalometer model for apportioning BC_ff and BC_bb. Although the authors provided references supporting their chosen values for this parameter, there are more recent studies regarding BC's AAE which show a quite wide range of 0.6-1.3 (Liu et al, ACP 18, 6259-6273, 2018). I would suggest to do an uncertainty and sensitivity analysis to investigate the impact of AAE on their results and provide a range of apportionment results instead of a certain value based on AAE = 1 or 2. Reference: Chao Liu, Chu Eddy Chung, Yan Yin, and Martin Schnaiter, The absorption Ångström exponent of black carbon: from numerical aspects, Atmos. Chem. Phys., 18, 6259–6273, 2018.

Response:

Thanks very much for the reviewer's valuable suggestion. We agree with the reviewer that AAE values for fossil fuel BC and biomass BC play an important role in Aethalometer model for apportioning $BC_{ff}$ and $BC_{bb}$.

One of the largest uncertainties for the Aethalometer model is related to the choice of the $\alpha_{ff}$ and $\alpha_{bb}$ values. The site-specific $\alpha$ values are affected by the type of fuel, combustion regime, and the mixing state of BC aerosols with non-absorbing materials. As stated in the reference (Liu et al., 2018) provided by the reviewer, AAE value of BC aerosol shows a quite wide range. This is also supported by many previous studies. Based on a comprehensive investigation on previous studies using the Aethalometer model (Table S1 in the revision), $\alpha_{ff}$ and $\alpha_{bb}$ values were most commonly in the range of 0.9–1.1 and 1.7–2.2. In this work, the empirical value of $\alpha_{ff}$ and $\alpha_{bb}$ is adopted as 1.0 and 2.0 following many previous studies (Kirchstetter et al., 2004; Favez et al., 2010; Crippa et al., 2013; Fuller et al., 2014; Crilley et al., 2015; Petit et al., 2017; Xiao et al., 2020). We implemented a sensitivity analysis according to the reviewer's suggestion do a sensitivity analysis to investigate the impact of AAE on their results (Line 140-155, 250-259).

Line 134: "470nm was selected as near-ultraviolet wavelength" According to light spectrum, 470nm is blue light (450nm-490nm) within the visible light spectrum. Near UV is 300-400nm.

Response:

We are sorry for this inaccurate statement. We have corrected the statement.

Line 140: $n_{\Delta\theta}$ is the total occurrences from wind sector $\Delta\theta$. Did you consider the duration of the occurrence? For example, there were two occurrences that the wind blew from 45 degree. One lasted for 1 sec and the other one lasted for 1 min, during which BC exceeded the threshold criterion. If you use occurrence in equation 5, the CPF is 50%. If you use duration in equation 5, the CFP is 98.4%. Which method do you think makes more sense?

Response:

We completely agree that the duration of the occurrence will affect the CPF results. In this study, we did not consider the duration of the occurrence. The CPF analysis were conducted with the average wind speed/direction with temporal resolution of 1 h. Therefore, we assumed that all the wind lasted for 1 h and they had the same occurrence.

Line 143: top 25% concentration was chosen as the threshold criterion. I did not see justification or reference for doing so.

Response:

The upper 25% was chosen as the threshold in order to be consistent with our previous study (Deng et al., 2020). The reference has been added in the revision.

Reference:

Deng, J., Zhao, W., Wang, X., Wu, L., Hu, W., Ren, L., and Fu, P.: Black carbon in Xiamen, China: temporal variations, transport pathways and impacts of synoptic circulation, *Chemosphere*, 241, 125133, 2020.

Lines 240-241: "BC_bb fraction is lowest at 8 am and INCREASE due to the decrease in traffic emission. If this is true, BC_bb should DECREASE when the traffic emission increase. Why at 19:00 rush hour BC_bb fraction reaches the highest instead of lowest, when the traffic emission peaks?

Response:

As shown in the figure (Fig. 5a), after 19:00, $BC_{bb}$ almost kept steady, while $BC_{ff}$ still increased gradually due to the enhanced traffic emissions from diesel trucks during nighttime. The heavy diesel trucks, which are major emission sources of $BC_{ff}$, were allowed to enter the city from 22:00 at late night to 7:00 in the early morning. Therefore, the $BC_{ff}$ fraction increased after 19:00 and accordingly the $BC_{bb}$ fraction reached its highest during the evening rush hour.

Lines 341-342: you see the discrepancy between the observation based results and modeling results and try to explain why modeling underestimate BC_bb in winter. It seems you have subjective preference for observation method over modeling method. As we know both methods have uncertainties and nobody really know what the true FF/BB apportionment is. In winter, Aethalometer method yielded larger BC_bb fraction than modeling method but it doesn't necessarily mean the modeling "underestimate" the BC_bb fraction. Could it be that the Aethalometer method "overestimate", or both overestimate but Aethalometer overestimate more, or both underestimate but Aethalometer underestimate less? I'd like to see more in-depth investigation regarding the discrepancy and an objective, comprehensive discussion of both observation and modeling results, instead of just focusing on the issues of modeling method.

Response:

We quite agree with you that both the Aethalometer method and the source-oriented model have uncertainties and therefore nobody really know what the "true" ff/bb apportionment is. The uncertainties in the two methods may lead to the gap between observation-based and model results. In our opinion, the uncertainties in the source apportionment methods come from many aspects.

For example, emission inventory used in the CMAQ model will significantly affect the performance of the model. Therefore, uncertainty in the emission inventory, including total amount and the spatiotemporal distribution, will lead to uncertainty in the model results. In addition, low resolution of simulation could also lead to uncertainties in model results. The peak values close to emission sources may not be captured after the dilution of emission in large grid cells. On the other hand, for the Aethalometer model, one of the largest uncertainties is related to the choice of the $\alpha_{ff}$ and $\alpha_{bb}$ values. We implemented a sensitivity analysis to

evaluate the impact of AAE selection on the apportionment results according to the reviewer's suggestion. The discussion about the gap between the two methods were implemented in the revision (Line 401-412).

---

## Author Response (AR1)

Author Responses to Referees' Comments on **"Source apportionment of black carbon aerosols from light absorption observation and source-oriented modeling: An implication in a coastal city in China"** by J. Deng et al. (MS No.: acp-2020-451)

5     We appreciate the valuable comments and suggestions from the reviewers. Our point-by-point responses are provided below. The referees' comments are in black and our responses are highlighted in red.

**Responses to Referee #1**

10    **General**

This paper presents a comparison and analysis of BC in the Xiamen region in China, a cleaner region than others from what I understand. The region is mostly affected by long range transport of pollutants. The authors have utilized light absorption measurements from the aethalometer (AE31-model) and further applied the source apportionment method denoted as the aethalometer model. Two BC fractions

15    originating from biomass burning (bb) and fossil fuel (ff) combustion where derived from the model and further compared with data from trajectory analysis, satellite observations and emission inventory modelling.

The novelty of the paper lies in the methodology, i.e. the comparison of the emission inventory model data to observations. Although, I don't agree with the authors that the model were in agreement with the

20    observations, the chosen methodology is conducted in a relatively clean region in China and pin-points the importance of comprehensive and correct emission inventories. The written language is on an average level. Some small errors occur occasionally which I haven't bothered to comment on. I would recommend this manuscript for publication in ACP after the following comments have been considered.

Response:

25    We would like to express our sincere thanks to the reviewer for the constructive and valuable comments, which helped us to effectively improve the quality of this paper. Besides, we accordingly revised some mistakes and improved the written English throughout the manuscript. Please see our itemized responses below.

30    **Major comments**

My largest concern with this paper is that the $BC_{ff}$ analysis and comparison to the CMAQ-model is lacking. A long piece of text is written to compare the $BC_{bb}$ to the CMAQ-model results, why is the $BC_{ff}$-comparison omitted?

Response:

35    In both the Aethalometer method and source-oriented CMAQ model, total BC was divided into two components, which were BC from fossil-fuel combustion (i.e. $BC_{ff}$) and biomass burning (i.e. $BC_{bb}$). If the observed $BC_{bb}$ contribution was smaller than the simulated $BC_{bb}$ contribution, as a matter of course, the observed $BC_{ff}$ contribution would be larger than the simulated $BC_{ff}$ contribution. And vice versa.

Therefore, in the original manuscript, we only compared $BC_{bb}$ contribution from the two methods and omitted the $BC_{ff}$-comparison.

According to this comment, the observed relative contribution of $BC_{ff}$ was implemented in Fig. 10, and $BC_{ff}$ analysis and comparison was also added in the revision.

The chosen AAE for the aethalometer model is very arbitrary selected. A lot has happened since the Sandradewi paper from 2008. Zotter et al. (2017) gave a comprehensive analysis of AAE selection. The authors may also consider conducting their own selection, perhaps based on a literature review.

Response:

Thanks for the valuable and helpful suggestion. In previous studies, fixed AAE values of fossil fuel ($\alpha_{ff}$) and biomass burning ($\alpha_{bb}$) were usually used in source apportionment studies for simplicity (Favez et al., 2010; Herich et al., 2011; Sciare et al., 2011; Harrison et al., 2013; Fuller et al., 2014; Rajesh and Ramachandran, 2017; Zotter et al., 2017; Helin et al., 2018). Therefore, one of the largest uncertainties in the Aethalometer model comes from the preselected $\alpha_{ff}$ and $\alpha_{bb}$ values (Sciare et al., 2011; Healy et al., 2017; Zotter et al., 2017; Helin et al., 2018). Zotter et al. (2017) determined the site specific $\alpha_{ff}$ and $\alpha_{bb}$ values by using auxiliary measurements (e.g. EC/OC, and $^{14}C$ measurements). Some other work estimated site specific $\alpha_{ff}$ and $\alpha_{bb}$ values using levoglucosan measurements together with the Aethalometer model (Fuller et al., 2014; Titos et al., 2017). In addition, many published studies commonly used values in the literature (Elser et al., 2016; Healy et al., 2017; Martinsson et al., 2017; Rajesh and Ramachandran, 2017; Xiao et al., 2020).

Based on a comprehensive literature review (Table S1 in the revision), we found that $\alpha_{ff}$ and $\alpha_{bb}$ values were most commonly in the range of 0.9–1.1 and 1.7–2.2, respectively. Due to the lack of supporting measurements (e.g. $^{14}C$ measurements, and chemical samples) in this work, the empirical value of $\alpha_{ff}$ and $\alpha_{bb}$ is adopted as 1.0 and 2.0 following many previous studies (Kirchstetter et al., 2004; Favez et al., 2010; Crippa et al., 2013; Fuller et al., 2014; Crilley et al., 2015; Petit et al., 2017; Xiao et al., 2020). The performance of the "Aethalometer model" is tested by sensitivity analysis using various combinations of AAE pairs for the assessment of $BC_{ff}$ and $BC_{bb}$. Therefore, except of the initial estimates with $\alpha_{ff}$=1.0 and $\alpha_{bb}$=2.0, the analysis was repeated by changing the $\alpha_{ff}$ from 0.9 to 1.1 and the $\alpha_{bb}$ from 1.7 to 2.2.

The above statements (Line 140-155) and the sensitivity analysis (Line 250-259) were added in the revision.

The authors claim that the CMAQ-model performed well and where in good agreement with the BC observations. How is this even possible to say? Studying Fig. 10, there is a considerable difference between the observations of $BC_{bb}$ and modeled $BC_{bb}$, especially during winter and spring. A statistical test, i.e. t-test or similar, would enable the authors to make a justified statement regarding the agreement. And, again the comparison to observed $BC_{ff}$ is lacking in Fig. 10 and totally omitted in the text.

Response:

We agree with the reviewer that there are considerable differences between the BC source apportionment results from the Aethalometer method and source-oriented CMAQ model in winter (14.3%) and spring (15.9%). We believe that the uncertainties in the Aethalometer method and the emission inventory used in the source-oriented model may lead to the unignorable gap between observation-based and model results. However, Fig. 10 also shows the discrepancies between the results from the two source apportionment methods are much smaller in summer (1.7%) and fall (6.0%). For the entire year, the annual average relative contributions of $BC_{ff}$ and $BC_{bb}$ derived by the source-oriented model are 67.4% and 32.6%, which are very close to those (i.e., 66.7% for $BC_{ff}$ and 33.3% for $BC_{bb}$) obtained by the Aethalometer method. In the revision, we accordingly revised our statements. In addition, we implemented the observed contribution of $BC_{ff}$ in Fig. 10, and added $BC_{ff}$ analysis and comparison in the revision based on the reviewer's suggestion.

**Minor comments**

L. 103. Please state the inlet of the AE31 as well as meters above sea level for the station.

Response:

The AE31 is with a $PM_{2.5}$ cut-off inlet. The station is ~10 m a.s.l. The statements have been revised (Line 105-110).

L. 115. Which MAC-values were used for the BC calculation?

Response:

The MAC value at 880 nm wavelength was 10.1 $m^2 g^{-1}$ based on our previous study (Deng et al., 2016).

Reference:

Deng, J., Zhang, Y., Hong, Y., Xu, L., Chen, Y., Du, W., and Chen, J.: Optical properties of PM2.5 and the impacts of chemical compositions in the coastal city Xiamen in China, *Sci. Total Environ.*, 557–558, 665–675, 2016.

L. 130-132. Selection of AAE, please see my comment above.

Response:

Thanks for this comment. Please see our response above.

L. 174. CMAQ is an abbreviation I suppose, for what?

Response:

CMAQ is the abbreviation for Community Multiscale Air Quality Modeling System. We have added it in the text (Line 196-197).

L. 216-218. The $PM_{2.5}$ was measured with which instrument?

Response:

Mass concentration of $PM_{2.5}$ was synchronously measured by a Tapered Element Oscillating Microbalance (TEOM) sampler (RP1400, Thermo Fisher Scientific). We have added the statement in

the revision (Line 262).

L. 216-219. This is a very strange sentence, which I don't understand with all these parentheses and all. Please rewrite.

Response:

We have rewritten the sentence as follows (Line 263-265):

"High-pollutant periods are the days with daily average concentration higher than the seasonal average plus one standard deviation, while low-pollutant periods are the days with daily average concentration lower than the seasonal average minus one standard deviation."

L. 215-225. Please explain clearly what you mean with high- and low-BC episodes.

Response:

Here, high-BC periods are the days with daily average BC concentration higher than the seasonal average plus one standard deviation (average + σ), and low-BC periods are the days with daily average BC concentration lower than the seasonal average minus one standard deviation (average - σ).

L. 220-225. Why not include Fig. S2 in the main manuscript?

Response:

Thanks for the suggestion. Accordingly, we moved Fig. S2 to the main manuscript as Fig. 4.

L. 239. How can $BC_{bb}$ be influenced by traffic?

Response:

The sentence has been revised to "$BC_{bb}$ was not influenced by traffic-related emission".

L. 241. How can the ratio be maximized when there is an evening rush-hour? I guess it is maximized due to increases in BB activity.

Response:

We agree with you that the maximum $BC_{bb}$ fraction toward evening is due to increase in biomass burning activity, such as domestic biofuel burning for cooking. We have revised the statement.

L. 253. I do understand that there is increased activity of domestic burning during winter. But why an increase in open-field biomass burning?

Response:

He et al. (2011) determined the seasonal variations in the open-filed biomass burning, including crop-residual burning and forest fire, by using the fire-counts data. They found that most of the fire hotspots around South China were detected in winter months since the period is typically on the harvest and dry season.

205    Response:

According to the suggestion, we added a similar analysis and brief comparison for $BC_{ff}$.

Figure 5. Please add more information in the Figure caption. The pie charts represents what? Why are there four trajectories. The four-coloured legend represents what? The percentage along the trajectories
210    represents what?

Response:

We revised the figure caption by adding more information according to the suggestion. In the figure, the pie charts represent relative contributions of $BC_{ff}$ (dark yellow) and $BC_{bb}$ (olive green). Four trajectory-clusters were obtained for each season with the clustering option of Euclidean distance.
215    Cluster analysis was performed with the TrajStat software and "Eye ball" method was used to determine the cluster number in this software. The four-coloured legend indicates the four different trajectory clusters. The percentage along the trajectories represent the percentage of each cluster in all trajectories.

Figure 10. Please include the fossil fuel combustion observations in this graph (i.e. $BC_{ff}$). There are no
220    Figure 10b as stated in the figure caption.

Response:

Thanks for the reminder. According the suggestion, we added the fossil fuel combustion observations into Fig. 10 and corrected the figure caption.

225

**Responses to Referee #2**

The manuscript by Deng at al. compared the source contributions from observation and CMAQ based source apportionment. Results indicate that while the BC predictions by the chemical transport model is an accordance with observations, a difference in source contributions can be observed. Overall, the manuscript is well written and addresses pertinent issues related to BC sources in a coastal city of China. To my knowledge this is one of the few studies which compare BC source apportionment using a source oriented model and observation based data. Such studies are needed as they directly evaluate the emission inventory of a region, with which many analysis and action plans to reduce air pollution are formulated.

Response:

We appreciate the reviewer's positive remarks on the importance of the work. We have revised the manuscript according to the reviewers' comments to improve its quality.

**Comments:**

1) As the observation data was collected in a single location, how reasonable it is to compare the results with a model simulation of 36X36 Km. In other words, why was a finer domain not selected? At least the authors should discuss the influence of grid resolution on the performance.

Response:

Although the observation showed in this study is from one single location, the model has been validated against all available observations within China including gas species as well as $PM_{2.5}$ and its components (Hu et al., 2015, 2016). A finer resolution is a good idea but is limited by lacking of input information such as emission inventory.

As suggested by the reviewer, we added below discussion regarding the grid resolution (Line 410-412): "In addition, low resolution of simulation could lead to uncertainties in model results. The peak values close to emission sources may not be captured after the dilution of emission in large grid cells. In future, high resolution is suggested as long as high-resolution emission inventories are available".

Response:

PSAT (i.e. Particulate matter Source Apportionment Technology) is similar to the sourced-oriented modeling with tagged tracers. However, PSAT does not track the species from different sources directly in each time step. Instead, it allocates the changes of bulk concentrations to different sources after each time step based on the ratio of each source to total emissions.

Comparing with the source-oriented model, the brute force method (BFM) is more suitable to estimate the change of PM due to proposed emission control measures than to determine the contributions of certain sources because removal of PM emissions could affect the transport, chemistry, deposition and interactions with meteorology although they are not chemically reactive (Zhang and Ying, 2011). The results simulated with the BFM is different from "source apportionment" since the summation of the contributions of all source categories will not always equal the total concentration. In addition, the BFM needs to repeat chemical transport model simulations multiple times and greatly increases the computational cost.

The differences of the source-oriented modeling and other methods such as brute force and PSAT have been clarified in the revision.

[Figure]

Figure R1. Correlations between (a) BCff and (b) BCbb calculated with the wavelength combinations of 470/950 nm and 370/880 nm.

4. How are $b_{abs,ff}$ and $b_{abs,bb}$ calculated? It's not clearly enough in the manuscript.

Response:

$b_{abs,ff}$ and $b_{abs,bb}$ are calculated with the method provided by Zotter et al. (2017).

The spectral dependence of the absorption is described by the power law $b_{abs}(\lambda) \sim \lambda^{-AAE}$. Consequently, for a wavelength pair the following relation can be derived:

$$\frac{b_{abs}(\lambda_1)}{b_{abs}(\lambda_2)} = \left(\frac{\lambda_1}{\lambda_2}\right)^{-\alpha} \tag{1}$$

In addition, $b_{abs}$ measured with the Aethalometer at different wavelengths can be apportioned into a biomass burning (bb) and a fossil fuel (ff) contribution. Therefore, $b_{abs,ff}$ and $b_{abs,bb}$ satisfy the following equation.

$$b_{abs}(\lambda) = b_{abs,ff}(\lambda) + b_{abs,bb}(\lambda) \tag{2}$$

Using Eqs. (1)-(2) and the measured babs at two different wavelengths, a fossil fuel and biomass burning contribution can be apportioned using the following equations:

$$\frac{b_{abs,ff}(\lambda_1)}{b_{abs,ff}(\lambda_2)} = \left(\frac{\lambda_1}{\lambda_2}\right)^{-\alpha_{ff}} \tag{3}$$

$$\frac{b_{abs,bb}(\lambda_1)}{b_{abs,bb}(\lambda_2)} = \left(\frac{\lambda_1}{\lambda_2}\right)^{-\alpha_{bb}} \tag{4}$$

$$b_{abs,bb} = \frac{b_{abs}(\lambda_1) - b_{abs}(\lambda_2)\cdot\left(\frac{\lambda_1}{\lambda_2}\right)^{-\alpha_{ff}}}{\left(\frac{\lambda_1}{\lambda_2}\right)^{-\alpha_{bb}} - \left(\frac{\lambda_1}{\lambda_2}\right)^{-\alpha_{ff}}} \tag{5}$$

$$b_{abs,ff} = \frac{b_{abs}(\lambda_1) - b_{abs}(\lambda_2)\cdot\left(\frac{\lambda_1}{\lambda_2}\right)^{-\alpha_{bb}}}{\left(\frac{\lambda_1}{\lambda_2}\right)^{-\alpha_{ff}} - \left(\frac{\lambda_1}{\lambda_2}\right)^{-\alpha_{bb}}} \tag{6}$$

For given $\alpha_{ff}$ and $\alpha_{bb}$ values and $b_{abs}(\lambda_1)$ and $b_{abs}(\lambda_2)$, the values for $b_{abs,ff}$ and $b_{abs,bb}$ can be calculated by combining Eqs. (5) and (6).

The details have been added in the revision.

**Minor comments**

1. L174: Define the abbreviation CMAQ when it appeared for the first time.

Response:

We have revised it in the revision (Line 196) accordingly.

2. L310: Why the South China Sea was source of BC?

Response:

Thanks for pointing out this mistake. In this study, the East China Sea and South China Sea were indicated as the potential source for BC. However, these areas should not be actual source areas and might be identified due to the trailing effect (Lee et al., 2014; Deng et al., 2020). We have corrected the statements in the revision (Line 361-363).

Response:

According to the reviewers' suggestion, we checked the manuscript throughout and corrected some spelling and grammar mistakes to improve its quality.

440

**Responses to Referee #4**

As BC aerosols play an important role in climate change and haze pollution, it has been a hot topic to
445    identify, source apportion, and locate their sources. The main methods include observation based receptor modeling, emission inventory, atmospheric transport simulation. This study integrated the observation based modeling and emission inventory based transportation modeling to provide a more comprehensive picture regarding BC sources in a southeast coastal city. It clearly showed advancement comparing to similar studies in this topic and merit publication. Below are some comments that I would
450    like the authors to address to improve its scientific quality.

Response:

Your positive comments are greatly appreciated. We have revised the manuscript to improve its quality according to the comments from the reviewers.

455    Lines 125 and 131: the authors assumed AAE values for fossil fuel BC and biomass BC (i.e., 1 and 2), which play an important role in Aethalometer model for apportioning BC_ff and BC_bb. Although the authors provided references supporting their chosen values for this parameter, there are more recent studies regarding BC's AAE which show a quite wide range of 0.6-1.3 (Liu et al, ACP 18, 6259-6273, 2018). I would suggest to do an uncertainty and sensitivity analysis to investigate the impact of AAE on
460    their results and provide a range of apportionment results instead of a certain value based on AAE = 1 or 2. Reference: Chao Liu, Chu Eddy Chung, Yan Yin, and Martin Schnaiter, The absorption Ångström exponent of black carbon: from numerical aspects, Atmos. Chem. Phys., 18, 6259–6273, 2018.

Response:

Thanks very much for the reviewer's valuable suggestion. We agree with the reviewer that AAE values
465    for fossil fuel BC and biomass BC play an important role in Aethalometer model for apportioning $BC_{ff}$ and $BC_{bb}$.

One of the largest uncertainties for the Aethalometer model is related to the choice of the $\alpha_{ff}$ and $\alpha_{bb}$ values. The site-specific $\alpha$ values are affected by the type of fuel, combustion regime, and the mixing state of BC aerosols with non-absorbing materials. As stated in the reference (Liu et al., 2018) provided
470    by the reviewer, AAE value of BC aerosol shows a quite wide range. This is also supported by many previous studies. Based on a comprehensive investigation on previous studies using the Aethalometer model (Table S1 in the revision), $\alpha_{ff}$ and $\alpha_{bb}$ values were most commonly in the range of 0.9–1.1 and 1.7–2.2. In this work, the empirical value of $\alpha_{ff}$ and $\alpha_{bb}$ is adopted as 1.0 and 2.0 following many

previous studies (Kirchstetter et al., 2004; Favez et al., 2010; Crippa et al., 2013; Fuller et al., 2014; Crilley et al., 2015; Petit et al., 2017; Xiao et al., 2020). We implemented a sensitivity analysis according to the reviewer's suggestion do a sensitivity analysis to investigate the impact of AAE on their results (Line 140-155, 250-259).

Line 134: "470nm was selected as near-ultraviolet wavelength" According to light spectrum, 470nm is blue light (450nm-490nm) within the visible light spectrum. Near UV is 300-400nm.

Response:

We are sorry for this inaccurate statement. We have corrected the statement.

Line 140: $n_{\Delta\theta}$ is the total occurrences from wind sector $\Delta\theta$. Did you consider the duration of the occurrence? For example, there were two occurrences that the wind blew from 45 degree. One lasted for 1 sec and the other one lasted for 1 min, during which BC exceeded the threshold criterion. If you use occurrence in equation 5, the CPF is 50%. If you use duration in equation 5, the CFP is 98.4%. Which method do you think makes more sense?

Response:

We completely agree that the duration of the occurrence will affect the CPF results. In this study, we did not consider the duration of the occurrence. The CPF analysis were conducted with the average wind speed/direction with temporal resolution of 1 h. Therefore, we assumed that all the wind lasted for 1 h and they had the same occurrence.

Line 143: top 25% concentration was chosen as the threshold criterion. I did not see justification or reference for doing so.

Response:

The upper 25% was chosen as the threshold in order to be consistent with our previous study (Deng et al., 2020). The reference has been added in the revision.

Reference:

Deng, J., Zhao, W., Wang, X., Wu, L., Hu, W., Ren, L., and Fu, P.: Black carbon in Xiamen, China: temporal variations, transport pathways and impacts of synoptic circulation, *Chemosphere*, 241, 125133, 2020.

Lines 240-241: "BC_bb fraction is lowest at 8 am and INCREASE due to the decrease in traffic emission. If this is true, BC_bb should DECREASE when the traffic emission increase. Why at 19:00 rush hour BC_bb fraction reaches the highest instead of lowest, when the traffic emission peaks?

Response:

As shown in the figure (Fig. 5a), after 19:00, $BC_{bb}$ almost kept steady, while $BC_{ff}$ still increased gradually due to the enhanced traffic emissions from diesel trucks during nighttime. The heavy diesel trucks, which are major emission sources of $BC_{ff}$, were allowed to enter the city from 22:00 at late night to 7:00 in the early morning. Therefore, the $BC_{ff}$ fraction increased after 19:00 and accordingly the $BC_{bb}$

fraction reached its highest during the evening rush hour.

Lines 341-342: you see the discrepancy between the observation based results and modeling results and try to explain why modeling underestimate BC_bb in winter. It seems you have subjective preference for observation method over modeling method. As we know both methods have uncertainties and nobody really know what the true FF/BB apportionment is. In winter, Aethalometer method yielded larger BC_bb fraction than modeling method but it doesn't necessarily mean the modeling "underestimate" the BC_bb fraction. Could it be that the Aethalometer method "overestimate", or both overestimate but Aethalometer overestimate more, or both underestimate but Aethalometer underestimate less? I'd like to see more in-depth investigation regarding the discrepancy and an objective, comprehensive discussion of both observation and modeling results, instead of just focusing on the issues of modeling method.

Response:

We quite agree with you that both the Aethalometer method and the source-oriented model have uncertainties and therefore nobody really know what the "true" ff/bb apportionment is. The uncertainties in the two methods may lead to the gap between observation-based and model results. In our opinion, the uncertainties in the source apportionment methods come from many aspects.

For example, emission inventory used in the CMAQ model will significantly affect the performance of the model. Therefore, uncertainty in the emission inventory, including total amount and the spatiotemporal distribution, will lead to uncertainty in the model results. In addition, low resolution of simulation could also lead to uncertainties in model results. The peak values close to emission sources may not be captured after the dilution of emission in large grid cells. On the other hand, for the Aethalometer model, one of the largest uncertainties is related to the choice of the $\alpha_{ff}$ and $\alpha_{bb}$ values. We implemented a sensitivity analysis to evaluate the impact of AAE selection on the apportionment results according to the reviewer's suggestion.

The discussion about the gap between the two methods were implemented in the revision (Line 401-412).

[revised manuscript text omitted]
 also characterizeddemonstrated in Fig. 11. and compared with relative contributions of biomass burning estimated according to light absorption properties (Fig. 10). The source oriented model almost captured the seasonal trends in contributions of fossil fuel combustion and biomass burning, although its largest seasonal contribution of biomass burning occurred in spring rather than winter. For the source oriented modeling resultsentire year, liquid fossil fuel combustion had the largest contribution (46.5 %), followed by biomass burning (32.6 %) and coal combustion (20.9 %). Contributions of different fuel catalogs exhibited distinct seasonality. Seasonal contributions of coal combustion were in the order of winter (27.8 %) > fall (23.4 %) > spring (15.6 %) > summer (14.5 %). Different from coal combustion and biomass burningFor, liquid fossil fuel combustion had the, its largest contribution (57.5 %) was in summer (57.5 %) whileand its smallest contribution (37.4 %) was in spring. Seasonal contributions of biomass burning were in the range of 25.7 % – 47.0 %, and the average contribution was much larger in spring than in other seasons. Contribution of fossil fuel combustion, which is the sum of liquid fossil fuel and coal combustion, followed the order of winter > fall > summer > spring.

The simulated contributions were compared with the source apportionment results estimated according to light-absorption properties. For the whole year, the annual average relative contributions of $BC_{ff}$ and $BC_{bb}$ derived by the source-oriented model were 67.4% and 32.6%, respectively. They were very close to the results (i.e., 66.7% for $BC_{ff}$ and 33.3% for $BC_{bb}$) obtained by the Aethalometer method. The overall consistence of the two apportionment methods confirms that the source apportionment results in Xiamen from this study are reasonable and benefiting future emission-control strategies. Simulated contributions of $BC_{ff}$ and $BC_{bb}$ were 72.0 % and 28.0 % in summer, and 73.9 % and 26.1 % in fall. The simulated contributions in summer and fall were comparable to those derived by the Aethalometer method, and the discrepancies between the results from the two methods were 1.7 % in summer and 6.0 % in fall. However, there were considerable differences between the BC source apportionment results from the Aethalometer method and source-oriented CMAQ model in winter and spring. Compared to observation based results, relative Relative contributions of $BC_{ff}$ and $BC_{bb}$biomass burning derived by the source-oriented modeling wereas 74.314.3 % and 25.7 %smaller in winter, while and 47.0 % and

15.953.0 % larger in spring. Simulated $BC_{ff}$ ($BC_{bb}$) contribution was 14.3 % larger (smaller) in winter and 15.9 % smaller (larger) in spring compared to the observation-based results. Both the uncertainties in the Aethalometer method and the emission inventory used in the source-oriented model may lead to the gap between observation-based and model results. For example, due to lack of the auxiliary measurements, the AAE values of $BC_{ff}$ and $BC_{bb}$ adopted in this study were based on a comprehensive literature review. It would lead to uncertainties in the apportionment results from the Aethalometer model, as discussed in Section 3.1. On the other hand, Tthe large gap in spring and winter may partly be resulted from the uncertainty uncertainties in satellite-based inventory of biomass-burning emissions in South Asia and south China (Wiedinmyer et al., 2011), while the disagreement in winter may ascribe to the underestimation of biomass burning emissions in north and central China (; Huang et al., 2012; Zhou et al., 2016). In China, some open burning activities such as local/small-scale open burning and smoldering are important sources of biomass-burning BC, which was particularly the case in winter. However, these burning activities are difficult to be accurately detected by satellite, leading to considerable underestimation ofuncertainties in biomass-burning emissions. The discrepancies between the results from the two source apportionment methods were much smaller in summer (1.7 %) and fall (6.0 %). Seasonal relative contributions of biomass burning estimated by the source oriented model were in the range of 25.7 % – 47.0 %, with an annual average of 32.6 %, which was very close to the observed results derived from the Aethalometer method. The consistence of the two different source 
[revised manuscript text omitted]

[Figure]

Figure 35. Diurnal and monthly variations in BC$_{ff}$ and BC$_{bb}$ concentrations with the BC$_{bb}$/BC fraction.

[Figure]

750

**Figure 46. CPF plots for (a) BC$_{ff}$, (b) BC$_{bb}$ and (c) BC$_{bb}$ contribution in Xiamen in 2014. ws represents wind speed (m s$^{-1}$).**

[Figure]

**Figure 5.** Seasonal cluster-mean of five-day backward trajectories at 500 m with the corresponding trajectory percentages and BC$_{bb}$ contributions in Xiamen. The pie charts represent relative contributions of BC$_{ff}$ (dark yellow) and BC$_{bb}$ (olive green). The four-colored legend indicates the four different trajectory clusters. The percentage along the trajectories represent the percentage of each cluster in all trajectories.

[Figure]

[Figure]

760

**Figure 68.** oncentration-weighted trajectory (CWT) maps (ng m$^{-3}$) for (ca) BC$_{ff}$ and (db) BC$_{bb}$ in Xiamen in 2014.

[Figure]

765    **Figure 89. Concentration weighted trajectory (CWT) maps for BCff and BCbb in Xiamen for different seasons in 2014.**

[Figure]

**Figure 10. Source contributions to BC of five source sectors in each period based on source-oriented model.**

[Figure]

770

[Figure]

**Figure** 11. seasonal  source contribution  to BC between source-oriented model and Aethalometer method.